# Recurrent Independent Mechanisms

## Abstract

Learning modular structures which reflect the dynamics of the environment can lead to better generalization and robustness to changes which only affect a few of the underlying causes. We propose Recurrent Independent Mechanisms (RIMs), a new recurrent architecture in which multiple groups of recurrent cells operate with nearly independent transition dynamics, communicate only sparingly through the bottleneck of attention, and are only updated at time steps where they are most relevant. We show that this leads to specialization amongst the RIMs, which in turn allows for dramatically improved generalization on tasks where some factors of variation differ systematically between training and evaluation.

## 1 Introduction

Physical processes in the world often have a modular structure, with complexity emerging through combinations of simpler subsystems. Machine learning seeks to uncover and use regularities in the physical world. Although these regularities manifest themselves as statistical dependencies, they are ultimately due to dynamic processes governed by physics. These processes are often independent and only interact sparsely. For instance, we can model the motion of two balls as separate independent mechanisms even though they are both gravitationally coupled to Earth as well as (weakly) to each other. They may, however, occasionally strongly interact via collisions.

The notion of independent or autonomous mechanisms has been influential in the field of causal inference, where it is applied not only to dynamic processes but also to time independent datasets. For instance, it has been argued that the conditional distribution of the average annual temperature given the altitude of a place is an abstraction of a causal mechanism (subsuming complex physical processes involving air pressure, etc.) that is independent of the distribution of the altitudes of settlements (Peters et al., 2017), and will thus apply invariantly for, say, different countries in the same climate zone with different altitude distributions.

A complex generative model, temporal or not, can be thought of as the composition of independent mechanisms or "causal" modules. In the causality community, this is often considered a prerequisite of being able to perform localized interventions upon variables determined by such models (Pearl, 2009). It has been argued that the individual modules tend to remain robust or invariant even as other modules change, e.g., in the case of distribution shift (Schölkopf et al., 2012; Peters et al., 2017). One may hypothesize that if a brain is able to solve multiple problems beyond a single i.i.d. (independent and identically distributed) task, it would be economical to learn structures aligned with this, by learning independent mechanisms that can flexibly be reused, composed and re-purposed.

In the dynamic setting, we think of an overall system being assayed as composed of a number of fairly independent subsystems that evolve over time, responding to forces and interventions. A learning agent then need not devote equal attention to all subsystems at all times: only those aspects that significantly interact need to be considered jointly when taking a decision or forming a plan (Bengio, 2017). Such sparse interactions can reduce the difficulty of learning since few interactions need to be considered at a time, reducing unnecessary interference when a subsystem is adapted. Models learned this way may be more likely to capture the compositional generative (or causal) structure of the world, and thus better generalize across tasks where a (small) subset of mechanisms change while most of them remain invariant (Simon, 1991; Peters et al., 2017; Parascandolo et al., 2018). The central question motivating our work is how a machine learning approach can learn independent but sparsely interacting recurrent mechanisms in order to benefit from such modularity.

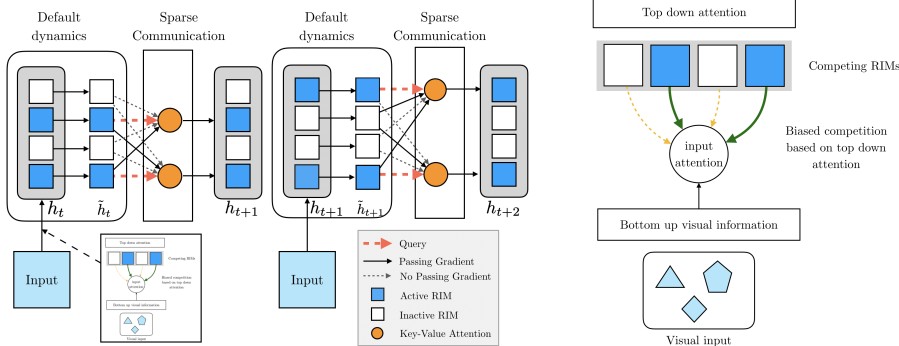

Figure 1: **Illustration of Recurrent Independent Mechanisms (RIMs)**. A single step under the proposed model occurs in four stages (left figure shows two steps). In the first stage, individual RIMs produce a query which is used to read from the current input. In the second stage, an attention based competition mechanism is used to select which RIMs to activate (right figure) based on encoded visual input (blue RIMs are active, based on attention score, white RIMs remain inactive). In the third stage, individual activated RIMs follow their own default transition dynamics while non-activated RIMs remain unchanged. In the fourth stage, the RIMs sparsely communicate information between themselves, also using attention.

## 2 RECURRENT INDEPENDENT MECHANISMS WITH SPARSE INTERACTIONS

Our approach to modelling a dynamical system of interest divides the overall model into $k$ small subsystems (or modules), each of which is recurrent in order to be able to capture dynamics. We refer to these subsystems as *Recurrent Independent Mechanisms (RIMs)*, where each RIM has distinct functions that are learned automatically from data[1]. We refer to RIM $k$ at time step $t$ as having state $h_{t,k}$, where $t = 1, \ldots, T$. Each RIM has parameters $\theta_k$, which are shared across all time steps.

At a high level (see. Fig. 1), we want each RIM to have its own independent dynamics operating by default, and occasionally to interact with other relevant RIMs and with selected elements of the encoded input. The total number of parameters can be kept small since RIMs can specialize on simple sub-problems, similar to Parascandolo et al. (2018). This specialization and modularization not only has computational and statistical advantages (Baum & Haussler, 1989; Bengio et al., 2019), but also prevents individual RIMs from dominating and modelling complex, composite mechanisms. We expect this to lead to more robust systems than training one big homogeneous neural network (Schmidhuber, 2018). Moreover, modularity also has the desirable implication that a RIM should maintain its own independent functionality even as other RIMs are changed. A more detailed account of the desiderata for the model is given in Appendix A.

### 2.1 INDEPENDENT RIM DYNAMICS

Now, consider the default transition dynamics which we apply for each RIM independently and during which no information passes between RIMs. We use $\tilde{h}$ for the hidden state after the independent dynamics are applied (and before attention is applied). First, for the RIMs which are not activated (we refer to the activated set as $\mathcal{S}_t$), the hidden state remains unchanged:

$$\tilde{h}_{t+1,k} = h_{t,k} \qquad \forall k \notin \mathcal{S}_t. \tag{1}$$

Note that the gradient still flows through a RIM on a step where it is not activated. For the RIMs that are activated, we run a per-RIM independent transition dynamics. The form of this is somewhat flexible, but in this work we opted to use either a GRU (Chung et al., 2015) or an LSTM (Hochreiter & Schmidhuber, 1997). We generically refer to these independent transition dynamics as $D_k$, and we emphasize that each RIM has its own separate parameters. Aside from being RIM-specific, the internal operation of the LSTM and GRU remain unchanged, and the active RIMs are updated by

$$\tilde{h}_{t+1,k} = D_k(h_{t,k}) = LSTM(h_{t,k}, A_k^{(in)}; \theta_k^{(D)}) \qquad \forall k \in \mathcal{S}_t \tag{2}$$

as a function of the attention mechanism $A_k^{(in)}$ applied on the current input, described in the next two subsections below, after explaining the key-value mechanism used to select arguments for this update.

---

[1] Note that we are using the term *mechanism* both for the mechanisms that make up the world's dynamics as well as for the computational modules that we learn to model those mechanisms.

## 2.2 Key-Value Attention to Process Sets of Named Interchangeable Variables

Each RIM should be activated and updated when the input is relevant to it. We thus utilize competition to allocate representational and computational resources. As argued by Parascandolo et al. (2018), this tends to produce independence among learned mechanisms, provided the training data has been generated by a set of independent physical mechanisms. In contrast to Parascandolo et al. (2018), we use an *attention mechanism* for this purpose. In doing so, we are inspired by findings from experimental psychology in the study of the interplay of top-down attention and bottom-up information flow, conceptualized in the *biased competition theory* of selective attention (Desimone & Duncan, 1995): A brain's capacity for parallel processing of complex entities is limited, and many brain systems representing visual information use competition (operating in parallel across the visual field) to allocate resources, often biased by feedback from higher brain areas.

The introduction of content-based soft-attention mechanisms (Bahdanau et al., 2014) has opened the door to neural networks which operate on *sets of typed interchangeable objects*. This idea has been remarkably successful and widely applied to most recent Transformer-style multi-head dot product self attention models (Vaswani et al., 2017; Santoro et al., 2018), achieving new state-of-the-art results in many tasks. Soft-attention uses the product of a *query* (or *read key*) $Q$ of dimensionality $N_r \times d$ matrix $Q$, and $d$ dimension of each key) to a set of $N_o$ objects each associated with a *key* (or *write-key*) matrix $K^T$ ($N_o \times d$), and after normalization with a softmax yields outputs in the convex hull of the *values* (or *write-values*) $V_i$ (row $i$ of matrix $V$). Its result is computed as

$$\text{Attention}(Q, K, V) = \text{softmax}\left(\frac{QK^T}{\sqrt{d}}\right)V,$$

where the softmax is applied to each row of its argument matrix, yielding a set of convex weights. As a result, one obtains a convex combination of the values $V$. If the attention is focused on one element for a particular row (i.e., the softmax is saturated), this simply selects one of the objects and copies its value to row $j$ of the result. Note that the $d$ dimensions in the key can be split into *heads* which then have their attention matrix and write values computed separately.

When the inputs and outputs of each RIM are a set of objects or entities (each associated with a key and value vector), the RIM processing becomes a generic object-processing machine which can operate on "variables" in a sense analogous to variables in a programming language: as interchangeable arguments of functions. Because each object has a key embedding (which one can understand both as a name and as a type), the same RIM processing can be applied to any variable which fits an expected "distributed type" (specified by a query vector). Each attention head then corresponds to a typed argument of the function computed by the RIM. When the key of an object matches the query, it can be used as input for the RIM. Whereas in regular neural networks (without attention) neurons operate on fixed variables (the neurons which are feeding them from the previous layer), the key-value attention mechanisms make it possible to select on the fly which variable instance (i.e. which entity or object) is going to be used as input for each of the arguments of the RIM dynamics, with a different set of query embeddings for each RIM. These inputs can come from the external input or from the output of other RIMs. So, if the individual RIMs can represent these "functions with typed arguments," then they can "bind" to whatever input is currently available and best suited according to its attention score: the "input attention" mechanism would look at the candidate input object's key and evaluate if its "type" matches with what this RIM expects (specified in the query).

## 2.3 Selective Activation of RIMs as a form of Top-Down Modulation

The proposed model learns to dynamically select those RIMs for which the current input is relevant. We give each RIM the choice between attending to the actual input instances or a special null input. The null input consists entirely of zeros and thus contains no information. At each step, we select the top-$k_A$ (out of $k_T$) RIMs in terms of their value of the softmax for the real input. Intuitively, the RIMs must compete on each step to read from the input, and only the RIMs that win this competition will be able to read from the input and have their state updated.

In our use of key-value attention, the queries come from the RIMs, while the keys and values come from the current input. The mechanics of this attention mechanism follow from the Transformer (Vaswani et al., 2017) and the RMC (Santoro et al., 2018), with the modification that the parameters of the attention mechanism itself are separate for each RIM. The input attention for a particular RIM

is described as follows. The input $x_t$ at time $t$ is seen as a set of elements, structured as rows of a matrix (for image data, it can be the output of the CNN). We first concatenate a row full of zeros, to obtain

$$X = \emptyset \oplus x_t. \tag{3}$$

$\oplus$ refers to the row-level concatenation operator. Then, linear transformations are used to construct keys ($K = XW^k$, one per input element and for the null element), values ($V = XW^v$, again one per element), and queries ($Q = RW_k^q$, one per RIM attention head) where $R$ is a matrix with each row $r_i$ corresponding to the hidden state of an individual RIM (i.e $h_{t,k}$). $W^v$ is a simple matrix mapping from an input element to the corresponding value vector for the weighted attention and $W^k$ is similarly a weight matrix which maps the input to the keys. $W_k^q$ is a per-RIM weight matrix which maps from the RIM's hidden state to its queries. The attention thus is

$$A_k^{(in)} = \text{softmax}\left(\frac{RW_k^q (XW^k)^T}{\sqrt{d_e}}\right) XW^v, \text{ where } \theta_k^{(in)} = (W_k^q, W^e, W^v). \tag{4}$$

Based on the softmax values in (4), we select the top $k_A$ RIMs (out of the total $K$ RIMs) to be activated for each step, which have the least attention on the null input (and thus put the highest attention on the input), and we call this set $\mathcal{S}_t$. Since the queries depend on the state of the RIMs, this enables individual RIMs to attend only to the part of the input that is relevant for that particular RIM, thus enabling selective attention based on a *top-down attention* process (see. Fig 1). In practice, we use multiheaded attention, and multi-headed attention doesn't change the essential computation, but when we do use it for input-attention we compute RIM activation by averaging the attention scores over the heads.

## 2.4 COMMUNICATION BETWEEN RIMS

Although the RIMs operate independently by default, the attention mechanism allows sharing of information among the RIMs. Specifically, we allow the activated RIMs to read from all other RIMs (activated or not). The intuition behind this is that non-activated RIMs are not related to the current input, so their value should not change. However they may still store contextual information that is relevant for activated RIMs. For this communication between RIMs, we use a residual connection as in (Santoro et al., 2018) to prevent vanishing or exploding gradients over long sequences.

$$Q_{t,k} = \tilde{W}_k^q \tilde{h}_{t,k}, \quad \forall k \in \mathcal{S}_t \tag{5}$$

$$K_{t,k} = \tilde{W}_k^e \tilde{h}_{t,k}, \quad \forall k \tag{6}$$

$$V_{t,k} = \tilde{W}_k^v \tilde{h}_{t,k}, \quad \forall k \tag{7}$$

$$h_{t+1,k} = \text{softmax}\left(\frac{Q_{t,k}(K_{t,:})^T}{\sqrt{d_e}}\right) V_{t,:} + \tilde{h}_{t,k} \quad \forall k \in \mathcal{S}_t, \text{ where } \theta_k^{(c)} = (\tilde{W}_k^q, \tilde{W}_k^e, \tilde{W}_k^v). \tag{8}$$

As in the Transformer and RMC Vaswani et al. (2017); Santoro et al. (2018), we use multiple heads (as well as input attention (as in Sec 2.3) by producing different sets of queries, keys, and values to compute a linear transformation for each head (different heads have different parameters).

## 2.5 VARIATIONS ON THE RIMS ARCHITECTURE

The RIMs architecture that we study is highly homogeneous and generally the only hyperparameters are the number of RIMs $K$ and how many RIMs are activated on each time step $K_A$. All of the datasets that we consider are temporal, yet there is a distinction between datasets where the input on each time step is highly structured (such as a video, where each time step is an image) and where this is not the case (such as language modeling, where each step is a word or character). In the former case, we can get further improvements by making the activation of RIMs not just sparse across time but also sparse across the (spatial) structure.

## 3 RELATED WORK

**Neural Turing Machine (NTM) and Relational Memory Core (RMC):** the NTM (Graves et al., 2014a) consists of a sequence of independent memory cells, and uses an attention mechanism while performing targeted read and write operations. This shares a key idea with RIMs: that input information should only impact a sparse subset of the memory by default, while keeping most of the memory unaltered. RMC (Santoro et al., 2018) uses a multi-head attention mechanism to share information between multiple memory elements. We encourage the RIMs to remain separate as much as possible, whereas Santoro et al. (2018) allow information between elements to flow on each step in an unconstrained way. Instead, each RIM has its own default dynamics, while in RMC, all the processes interact with each other.

**Separate Recurrent Models:** EnTNet (Henaff et al., 2016) and IndRNN (Li et al., 2018) can be viewed as a set of separate recurrent models. In IndRNN, each recurrent unit has completely independent dynamics, whereas EntNet uses an independent gate for writing to each memory slot. RIMs use different recurrent models (with separate parameters), but we allow the RIMs to communicate with each other sparingly using an attention mechanism.

**Modularity and Neural Networks**: A neural network is composed of several neural modules, where each module is meant to perform a distinct function, and hence can be seen as a combination of experts (Jacobs et al., 1991; Bottou & Gallinari, 1991; Ronco et al., 1997; Reed & De Freitas, 2015; Andreas et al., 2016; Parascandolo et al., 2018; Rosenbaum et al., 2017; Fernando et al., 2017; Shazeer et al., 2017; Kirsch et al., 2018; Rosenbaum et al., 2019) routing information through a gated activation of layers. These works generally assume that only a single expert is active at a particular time step. In the proposed method, multiple RIMs can be active, interact and share information.

**Computation on demand:** There are various architectures (El Hihi & Bengio, 1996; Koutnik et al., 2014; Chung et al., 2016; Neil et al., 2016; Jernite et al., 2016; Krueger et al., 2016) where parts of the LSTM's hidden state are kept dormant at times. The major differences as compared to the proposed architecture are that (a) we modularize the dynamics of recurrent cells (using RIMs), and (b) we also control the inputs of each module (using transformer style attention), while many previous gating methods did not control the inputs of each module, but only whether they should be executed or not.

## 4 EXPERIMENTS

The main goal of our experiments is to show that the use of RIMs improves generalization across changing environments and/or in modular tasks, and to explore how it does so. Our goal is not to outperform highly optimized baselines; rather, we want to show the versatility of our approach by applying it to a range of diverse tasks, focusing on tasks that involve a changing environment. We organize our results by the capabilities they illustrate: we address generalization based on temporal patterns, based on objects, and finally consider settings where both of these occur together.

### 4.1 RIMS IMPROVE GENERALIZATION BY SPECIALIZING OVER TEMPORAL PATTERNS

We first show that when RIMs are presented with sequences containing distinct temporal patterns, they are able to specialize so that different RIMs are activated on different patterns. As a result, RIMs are able to generalize well when we modify a subset of the patterns (especially those unrelated to the class label) while most recurrent models fail to generalize well to these variations.

### 4.1.1 COPYING TASK

First we turn our attention to the task of receiving a short sequence of characters, then receiving blank inputs for a large number of steps, and then being asked to reproduce the original sequence. We can think of this as consisting of two temporal patterns which are independent: one where the sequence is received and another "dormant" pattern where no input is provided.

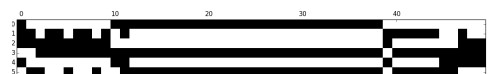

Figure 2: **Copying Task RIM Activation Pattern** for a model with $K = 6$ RIMs and $K_A = 3$ active RIMs per step (the activated RIMs are in black, non-activated in white). We can see that the RIM activation pattern is distinct during the dormant part of the sequence.

| Copying | $k_T$ | $k_A$ | $h_{size}$ | Train(50) CE | Test(200) CE |
|---|---|---|---|---|---|
| RIMs | 6 | 5 | 600 | 0.01 | 3.5 |
|  | 6 | 4 | 600 | **0.00** | **0.00** |
|  | 6 | 3 | 600 | **0.00** | **0.00** |
|  | 6 | 2 | 600 | **0.00** | **0.00** |
|  | 5 | 3 | 500 | **0.00** | **0.00** |
| LSTM | - | - | 300 | 0.00 | 2.28 |
|  | - | - | 600 | 0.00 | 3.56 |
| NTM | - | - | - | 0.00 | 2.54 |
| RMC | - | - | - | 0.00 | 0.13 |
| Transformers | - | - | - | 0.00 | 0.54 |

| Sequential MNIST | $k_T$ | $k_A$ | $h_{size}$ | 16 x 16 Accuracy | 19 x 19 Accuracy | 24 x 24 Accuracy |
|---|---|---|---|---|---|---|
| RIMs | 6 | 6 | 600 | 85.5 | 56.2 | 30.9 |
|  | 6 | 5 | 600 | 88.3 | 43.1 | 22.1 |
|  | 6 | 4 | 600 | **90.0** | **73.4** | **38.1** |
| LSTM | - | - | 300 | 86.8 | 42.3 | 25.2 |
|  | - | - | 600 | 84.5 | 52.2 | 21.9 |
| EntNet | - | - | - | 89.2 | 52.4 | 23.5 |
| RMC | - | - | - | 89.58 | 54.23 | 27.75 |
| DNC | - | - | - | 87.2 | 44.1 | 19.8 |
| Transformers | - | - | - | **91.2** | 51.6 | 22.9 |

Table 1: Performance on the copying task (left) and sequential MNIST resolution task right). **Error (CE on the last 10 time steps) on the copying task**. Note that while all of the methods are able to learn to copy for the length seen during training, the RIMs model generalizes to sequences longer than those seen during training whereas the LSTM, RMC, and NTM degrade. **Sequential MNIST resolution:** Test Accuracy % on the Sequential MNIST resolution generalization task (see text) after 100 epochs. Both the proposed and the Baseline model (LSTM) were trained on 14x14 resolution but evaluated at different resolutions; results averaged over 3 different trials.

As an example of out-of-distribution generalization, we find that using RIMs, we can extend the length of this dormant phase from 50 during training to 200 during testing and retain perfect performance (Table 1), whereas baseline methods including LSTM, NTM, and RMC substantially degrade. In addition, we find that this result is robust to the number of RIMs used as well as to the number of RIMs activated per-step. Our results (Appendix C.5) show that communication between different RIMs as well as input attention is necessary to achieve good generalization. We consider this preliminary evidence that RIMs can specialize over distinct patterns in the data and improve generalization to settings where these patterns change.

### 4.1.2 SEQUENTIAL MNIST RESOLUTION TASK

RIMs are motivated by the hypothesis that generalization performance can be improved by having modules which only activate on relevant parts of the sequence. For further evidence that RIMs can achieve this out-of-distribuution, we consider the task of classifying MNIST digits as sequences of pixels (Krueger et al., 2016) and assay generalization to images of resolutions different from those seen during training. Our intuition is that the RIMs model should have distinct subsets of the RIMs activated for pixels with the digit and empty pixels. As a result, RIMs should generalize better to greater resolutions by keeping the RIMs which store pixel information dormant over the empty regions of the image.

**Results:** Table 1 shows the result of the proposed model on the Sequential MNIST Resolution Task. If the train and test sequence lengths agree, both models achieve comparable test set performance. However, the RIMs model was relatively robust to changing the sequence length (by changing the image resolution), whereas the LSTM performance degraded more severely. This can be seen as a more involved analogue of the copying task, as MNIST digits contain large empty regions. It is essential that the model be able to store information and pass gradients through these regions. The RIMs outperform strong baselines such as Transformers, EntNet, RMC, as well as the Differentiable Neural Computer (DNC) (Graves et al., 2016).

### 4.2 RIMS LEARN TO SPECIALIZE OVER OBJECTS AND GENERALIZE BETWEEN THEM

We have presented evidence that RIMs can specialize over temporal patterns. We now turn our attention to showing that RIMs can specialize to objects, and show improved generalization to settings where we add or remove objects at test time.

### 4.2.1 BOUNCING BALL ENVIRONMENT

We consider a synthetic "bouncing balls" task in which multiple balls (of different masses and sizes) move using basic Newtonian physics (Van Steenkiste et al., 2018). What makes this task particularly suited to RIMs is that the balls move independently most of the time, except when they collide. During training, we predict the next frame at each time step using teacher forcing (Williams & Zipser, 1989). We can then use this model to generate multi-step rollouts.

As a preliminary experiment, we train on sequences of length 51 (the previous standard), using a binary cross entropy loss when predicting the next frame. We consider LSTM as baseline. We then produce rollouts, finding that RIMs are better able to predict future motion (examples in Figure 3, Figure 10 in Appendix and quantitative comparisons in Figure 4).

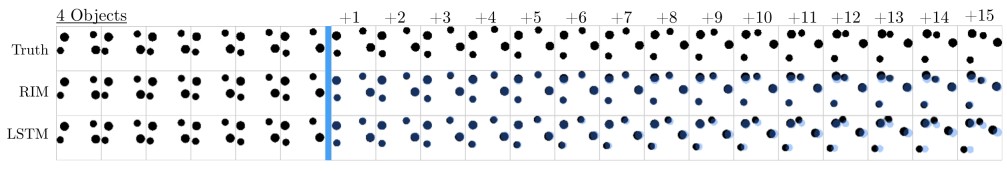

Figure 3: **Predicting Movement of Bouncing Balls**. The first 15 frames of ground truth are given (last 6 of those shown) and then the system is rolled out for the next 15 time steps. We find that RIMs perform better than the LSTMs (predictions are in black, ground truth in blue). Notice the blurring of LSTM predictions.

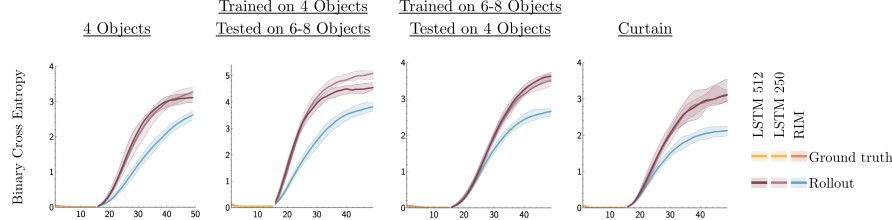

Figure 4: **Handling Novel Out-of-Distribution Variations**. Here, we study the performance of our proposed model compared to an LSTM baseline. The first 15 frames of ground truth are fed in and then the system is rolled out for the next 10 time steps. During the rollout phase, RIMs perform better than the LSTMs in accurately predicting the dynamics of the balls as reflected by the lower Cross Entropy (CE) [see blue for RIMs, purple for LSTM]. Notice the substantially better out-of-distribution generalization of RIMs when testing on a different number of objects than during training.

We take this further by evaluating RIMs on environments where the setup is different from the training setup. First we consider training with 4 balls and evaluating on an environment with 6-8 balls. Second, we consider training with 6-8 balls and evaluating with just 4 balls. Robustness in these settings requires a degree of invariance w.r.t. the number of balls.

In addition, we consider a task where we train on 4 balls and then evaluate on sequences where part of the visual space is occluded by a "curtain". This allows us to assess the ability of balls to be tracked (or remembered) through the occluding region. Our experimental results on these generalization tasks (Figure 4) show that RIMs substantially improve over an LSTM baseline. We found that increasing the capacity of the LSTM from 256 to 512 units did not substantially change the performance gap, suggesting that the improvement from RIMs is not primarily a result of increased capacity.

### 4.2.2 ENVIRONMENT WITH NOVEL DISTRACTORS

We next consider an object-picking reinforcement learning task from BabyAI (Chevalier-Boisvert et al., 2018) in which an agent must retrieve a specific object in the presence of distractors. We use a partially observed formulation of the task, where the agent only sees a small number of squares ahead of it. These tasks are difficult to solve (Chevalier-Boisvert et al., 2018) with standard RL algorithms, due to (1) the partial observability of the environment and (2) the sparsity of the reward, given that the agent receives a reward only after reaching the goal. During evaluation, we introduce new distractors to the environment which were not observed during training.

Figure 5 shows that RIMs outperform LSTMs on this task (details in appendix). When evaluating with known distractors, the RIM model achieves perfect performance while the LSTM struggles. When evaluating in an environment with novel unseen distractors the RIM doesn't achieve perfect performance but still outperforms the LSTM. An LSTM with a single memory flow may struggle to keep the distracting elements separate from elements which are necessary for the task, while the RIMs model uses attention to control which RIMs receive information at each step as well as what information they receive (as a function of their hidden state). This "top-down" bias results in a diminished representation of the distractor, not only enhancing the target visual information, but also suppressing irrelevant information. The notion that enhancement of the relevant information necessarily results in suppression of irrelevant information is fundamental to biased competition theory (Desimone & Duncan, 1995).

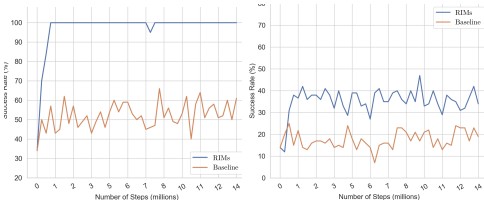

Figure 5: **Robustness to Novel Distractors:**. Left: performance of the proposed method compared to an LSTM baseline in solving the object picking task in the presence of distractors. Right: performance of proposed method and the baseline when novel distractors are added.

### 4.3 RIMs IMPROVE GENERALIZATION IN COMPLEX ENVIRONMENTS

We have investigated how RIMs use specialization to improve generalization to changing important factors of variation in the data. While these improvements have often been striking, it raises a question: what factors of variation should be changed between training and evaluation? One setting where factors of variation change naturally is in reinforcement learning, as the data received from an environment changes as the agent learns and improves. We conjecture that when applied to reinforcement learning, an agent using RIMs may be able to learn faster as its specialization leads to improved generalization to previously unseen aspects of the environment.

To investigate this we use an RL agent trained using Proximal Policy Optimization (PPO) (Schulman et al., 2017) with a recurrent network producing the policy. We employ an LSTM as a baseline, and compare results to the RIMs architecture. This was a simple drop-in replacement and did not require changing any of the hyperparameters for PPO. We experiment on the whole suite of Atari games and find that simply replacing the LSTM with RIMs greatly improves performance (Figure 6).

There is also an intriguing connection between the selective activation in RIMs and the concept of affordances from cognitive psychology (Gibson, 1977; Cisek & Kalaska, 2010). To perform well in environments with a dynamic combination of risks and opportunities, an agent should be ready to adapt immediately, releasing into execution actions which are at least partially prepared. This suggests agents should process sensory information in a contextual manner, building representations of potential actions that the environment currently affords. For instance, in Demon Attack, one of the games where RIMs exhibit strong performance gains, the agent must quickly choose between targeting distant aliens to maximize points and avoiding fire from close-by aliens to avoid destruction (indeed both types of aliens are always present, but which is relevant depends on the player's position). We hypothesize that in cases like this, selective activation of RIMs allows the agent to rapidly adapt its information processing to the types of actions relevant to the current context.

### 4.4 ABLATIONS

**Role of Top-Down Modulation: Removing Input Attention** We study the scenario where we remove the input attention process (Section 2.3) but still allow communication between RIMs (Section 2.4). We train this agent on 30 ATARI games for 30M time-steps each and compare the performance of this agent with the normal RIMs-PPO agent. We find that the RIMs agent still outperform this agent on 11 out of 30 games, while on 1 game (Frostbite) we see the proposed baseline agent substantially improves the performance. For more details regarding the training curves, refer to Fig. 25 (in Appendix).

**Importance of communication between RIMs:** For copying, we performed an ablation where we remove the communication between RIMs. We also varied the number of RIMs as well as the

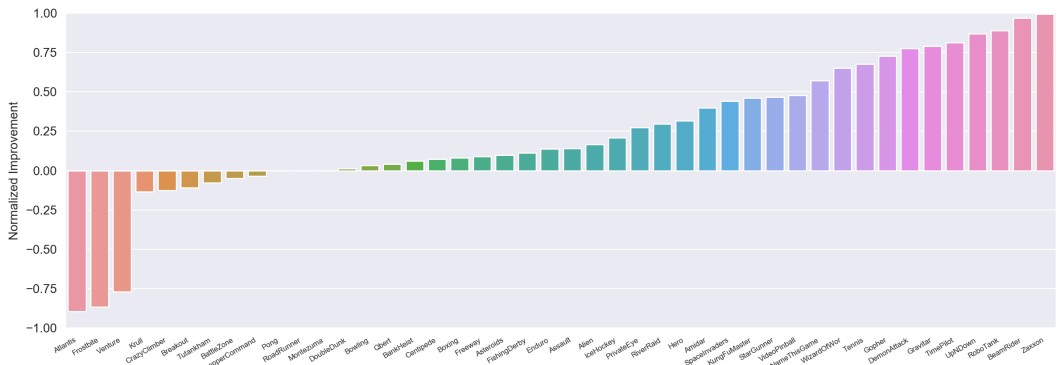

Figure 6: RIMs-PPO relative score improvement over LSTM-PPO baseline (Schulman et al., 2017) across all Atari games averaged over 3 trials per game. In both cases, PPO was used with the exact same settings, and the only change is the choice of recurrent architecture. More detailed experiments with learning curves are in Appendix C.

number of activated RIMs (Table 5). We found that the communication between RIMs is essential for good performance. We found similar results for the sequential MNIST resolution task.

**Importance of sparsity of activation of the RIMs** For the copying task, as well as for the sequential MNIST changed resolution task, we performed an ablation where we kept all RIMs active for all time steps (Table 5). We found that we were not able to achieve strong generalization as compared to the best performing RIMs model. On Atari we found that using $k_A = 5$ slightly improved over results compared with $k_A = 4$, but both had similar performance across the vast majority of games, suggesting that the $k_A$ hyperparameter is reasonably flexible in practice.

**Varying the number of attention heads for communication:** Here, we study what happens if the output of RIMs only has one 'object' rather than multiple ones (Section 2.2). The intuition is that RIM processing can be applied to any "head" which matches the query by an individual RIM. So, having more heads should help, as different heads could be used by different RIMs, rather than every RIM competing for the same head. We study this in the context of bouncing balls. We found that using multiple heads improves the performance, thus validating our hypothesis (Sec. 2.2). See Appendix C.11 for details.

**Randomly Dropping Out RIMs:** Modular structures are aggregates of mechanisms that can perform functions without affecting the remainder of the system, and interact as needed. To what extent are trained RIMs able to model meaningful phenomena when other RIMs are removed? We performed an experiment on moving MNIST digits where we train normally and "dropout" a random RIM at test time. We found that in the absence of selective activation (i.e. when $k_A = k_T$, Section C.13) the performance degraded very badly, but the performance degrades much less with selective activation. See Appendix C.13 for details.

## 5 CONCLUSION

Many systems of interest comprise multiple dynamical processes that operate relatively independently and only occasionally have meaningful interactions. Despite this, most machine learning models employ the opposite inductive bias, i.e., that all processes interact. This can lead to poor generalization (if data is limited) and lack of robustness to changing task distributions. We have proposed a new architecture, Recurrent Independent Mechanisms (RIMs), in which we learn multiple recurrent modules that are independent by default, but interact sparingly. Our positive experimental results lend support to the consciousness prior (Bengio, 2017), i.e., the importance of computational elements which focus on few mechanisms at a time in order to determine how a high-level state evolves over time, with many aspects of the state not being affected by this attentive dynamics (i.e., following default dynamics). For the purposes of this paper, we note that the notion of RIMs is not limited to the particular architecture employed here. The latter is used as a vehicle to assay and validate our overall hypothesis (cf. Appendix A), but better architectures for the RIMs model can likely be found.

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

# A    DESIDERATA FOR RECURRENT INDEPENDENT MECHANISMS

We have laid out a case for building models composed of modules which by default operate independently and can interact in a limited manner. Accordingly, our approach to modelling the dynamics of the world starts by dividing the overall model into small subsystems (or modules), referred to as *Recurrent Independent Mechanisms (RIMs)*, with distinct functions learned automatically from data.Our model encourages sparse interaction, i.e., we want most RIMs to operate independently and follow their default dynamics most of the time, only rarely sharing information. Below, we lay out desiderata for modules to capture modular dynamics with sparse interactions.

**Competitive Mechanisms:**    Inspired by the observations in the main paper, we propose that RIMs utilize competition to allocate representational and computational resources. As argued by (Parascandolo et al., 2018), this tends to produce independence among learned mechanisms if the training data has been generated by independent physical mechanisms.

**Top Down Attention:**    The points mentioned in Section 2 in principle pertain to synthetic and natural intelligent systems alike. Hence, it is not surprising that they also appear in neuroscience. For instance, suppose we are looking for a particular object in a large scene, using limited processing capacity. The *biased competition theory* of selective attention conceptualizes basic findings of experimental psychology and neuroscience (Desimone & Duncan, 1995): our capacity of parallel processing of and reasoning with high-level concepts is limited, and many brain systems representing visual information use competition to allocate resources. Competitive interactions among multiple objects occur automatically and operate in parallel across the visual field. Second, the principle of selectivity amounts to the idea that a perceiver has the ability to filter out unwanted information and selectively *process* the rest of the information. Third, *top-down bias* originating from higher brain areas enables us to selectively devote resources to input information that may be of particular interest or relevance. This may be accomplished by units matching the internal model of an object or process of interest being pre-activated and thus gaining an advantage during the competition of brain mechanisms.

**Sparse Information Flow:**    Each RIMs' dynamics should only be affected by RIMs which are deemed relevant. The fundamental challenge is centered around establishing sensible communication between RIMs. In the presence of noisy or distracting information, a large subset of RIMs should stay dormant, and not be affected by the noise. This way, training an ensemble of these RIMs can be more robust to out-of-distribution or distractor observations than training one big homogeneous neural network (Schmidhuber, 2018).

**Modular Computation Flow and Modular Parameterization:**    Each RIM should have its own dynamics operating by *default*, in the absence of interaction with other RIMs. The total number of parameters (i.e. weights) can be reduced since the RIMs can specialize on simple sub-problems, similar to (Parascandolo et al., 2018). This can speed up computation and improve the generalisation ability of the system (Baum & Haussler, 1989). The individuals RIMs in the ensemble should be simple also to prevent individual RIMs from dominating and modelling complex, composite mechanisms. We refer to a parameterization as modular if most parameters are associated to individuals RIMs only. This has the desirable property that a RIM should maintain its own independent functionality even as other RIMs are changed (due to its behavior being determined by its own self-contained parameters).

# B    EXTENDED RELATED WORK

Table 2: A concise comparison of recurrent models with modular memory.

| Method / Property | Modular Memory | Sparse Information Flow | Modular Computation Flow | Modular Parameterization |
|---|---|---|---|---|
| LSTM / RNN | ✗ | ✗ | ✗ | ✗ |
| Relational RNN (Santoro et al., 2018) | ✓ | ✗ | ✓ | ✗ |
| NTM (Graves et al., 2014b) | ✓ | ✓ | ✗ | ✗ |
| SAB(Ke et al., 2018) | ✗ | ✓ | ✗ | ✗ |
| IndRNN(Li et al., 2018) | ✓ | ✗ | ✗ | ✓ |
| RIMs | ✓ | ✓ | ✓ | ✓ |

The present section provides further details on related work, thus extending Section 3.

**Neural Turing Machine (NTM)**. The NTM (Graves et al., 2014a) has a Turing machine inspired memory with a sequence of independent memory cells, and uses an attention mechanism to move heads over the cells while performing targeted read and write operations. This shares a key idea with RIMs: that input information should only impact a sparse subset of the memory by default, while keeping most of the memory unaltered. The RIM

model introduces the idea that each RIM has its own independent dynamics, whereas the mechanism for updating memory cells update is shared.

**Relational RNN**. The Relational Models paper (Santoro et al., 2018) is based on the idea of using a multi-head attention mechanism to share information between multiple parts of memory. It is related to our idea but a key difference is that we encourage the RIMs to remain separate as much as possible, whereas (Santoro et al., 2018) allows information between the parts to flow on each step (in effect making the part distribution only relevant to a particular step). Additionally, RIMs has the notion of each RIM having its own independent transition dynamics which operate by default, whereas the Relational RNN only does computation and updating of the memory using attention.

**Sparse Attentive Backtracking (SAB)**. The SAB architecture (Ke et al., 2018) explores RNNs with self-attention across time steps as well as variants where the attention is sparse in the forward pass and where the gradient is sparse in the backward pass. It shares the motivation of using sparse attention to keep different pieces of information separated, but differs from the RIMs model in that it considers separation between time steps rather than separation between RIMs.

**Independently Recurrent Neural Network (IndRNN)**. The IndRNN (Li et al., 2018) replaces the full transition matrix in a vanilla RNN (between time steps) to a diagonal transition weight matrix. In other words, each recurrent unit has completely independent dynamics. Intriguingly they show that this gives much finer control over the gating of information, and allows for such an RNN to learn long-term dependencies without vanishing or exploding gradients. Analysis of the gradients shows that having smaller recurrent transition matrices mitigates the vanishing and exploding gradient issue. This may provide further explanation for why RIMs perform well on long sequences.

**Consciousness Prior** (Bengio, 2017): This is based on the assumption of a sparse graphical model describing the interactions between high-level variables, using gating mechanisms to select only a subset of high-level variables to interact at any particular time. This is closely related to our work in the sense high level abstract representation is based on the representations of the RIMs, which are activated sparsely and interact sparsely. Our paper thus helps to validate the consciousness prior idea.

**Recurrent Entity Networks**: EnTNet (Henaff et al., 2016) can be viewed as a set of separate recurrent models whose hidden states store the memory slots. These hidden states are either fixed by the gates, or modified through a simple RNN-style update. Moreover, EntNet uses an independent gate for writing to each memory slot. Our work is related in the sense that we also have different recurrent models (i.e.,RIMs, though each RIM has different parameters), but we allow the RIMs to communicate with each other sparingly using an attention mechanism.

**Capsules and Dynamic Routing:** EM Capsules (Hinton et al., 2018) and the preceding Dynamic Capsules (Sabour et al., 2017) use the poses of parts and learned part $\rightarrow$ object relationships to vote for the poses of objects. When multiple parts cast very similar votes, the object is assumed to be present, which is facilitated by an interactive inference (routing) algorithm.

**Relational Graph Based Methods:** Recent graph-based architectures have studied combinatorial generalization in the context of modeling dynamical systems like physics simulation, multi-object scenes, and motion-capture data, and multiagent systems (Scarselli et al., 2008; Bronstein et al., 2017; Watters et al., 2017; Raposo et al., 2017; Santoro et al., 2017; Gilmer et al., 2017; Van Steenkiste et al., 2018; Kipf et al., 2018; Battaglia et al., 2018; Tacchetti et al., 2018). One can also view our proposed model as a relational graph neural network, where nodes are parameterized as individual RIMs and edges are parameterized by the attention mechanism. Though, its important to emphasize that the topology of the graph induced in the proposed model is dynamic, while in most graph neural networks the topology is fixed.

**Default Behaviour:** Our work is also related to work in behavioural research that deals with two modes of decision making (Dickinson, 1985; Botvinick & Braver, 2015; Kool & Botvinick, 2018): an automatic systems that relies on habits and a controlled system that uses some privileged information for making decision making. The proposed model also has two modes of input processing, RIMs which activate uses some external sensory information, and hence analogous to controlled system. RIMs which don't activate, they are synonymous to habit based system. There is some work done trying in Reinforcement learning, trying to learn *default policies*, which have shown to improve transfer and generalization in multi-task RL (Teh et al., 2017; Goyal et al., 2019a). The proposed method is different in the sense, we are not trying to learn *default policies* which effect the environment, instead we want to learn mechanisms, which try to understand the environment. State dependent activation of different primitive policies was also studied in (Goyal et al., 2019b), and the authors showed that they can learn different primitives, but they also consider that only a single primitive can be active at a particular time step. Also, note that primitive policies try to *effect* the environment, whereas mechanism try to *understand* the enviornment.

## C  EXPERIMENTAL DETAILS AND HYPERPARAMETERS

### C.1  RIMS IMPLEMENTATION

The RIMs model consists of three main components: the input attention, the process for selecting activated RIMs, and the communication between RIMs. The input attention closely follows the attention mechanism of (Santoro et al., 2018) but with a significant modification: that all of the weights within the attention mechanism are separate per-block. Thus we remove the normal linear layers and replace them with a batch matrix multiplication over the RIMs (as each block has its own weight matrix). Note that the read-key (or query) is a function of the hidden state of each RIM.

For selecting activated RIMs, we compute the top-k attention weight on the null input over the RIMs. We then select the activated RIMs by using a mask.

We compute the independent dynamics over all RIMs by using a separate LSTM for each RIM. Following this, we compute the communication between RIMs as a multihead attention (Santoro et al., 2018), with the earlier-discussed modification of having separate weight parameters for each block, and also that we added a skip-connection around the attention mechanism. This attention mechanism used 4 heads and in general used a key size and value size of 32. We computed the updates for all RIMs but used the activated-block mask to selectively update only the activated subset of the RIMs.

The use of RIMs introduces two additional hyperparameters over an LSTM/GRU: the number of RIMs and the number of activated RIMs per step. We also observed that having too few activated RIMs tends to hurt optimization and having too many activated RIMs attenuates the improvements to generalization. For the future it would be interesting to explore dynamic ways of controlling how many RIMs to activate.

### C.2  DETAILED MODEL HYPERPARAMETERS

Table 3 lists the different hyperparameters.

Table 3: Hyperparameters

| Parameter | Value |
|---|---|
| Optimizer | Adam(Kingma & Ba, 2014) |
| learning rate | $7 \cdot 10^{-4}$ |
| batch size | 64 |
| Inp keys | 64 |
| Inp Values | Size of individual RIM * 4 |
| Inp Heads | 4 |
| Inp Dropout | 0.1 |
| Comm keys | 32 |
| Comm Values | 32 |
| Comm heads | 4 |
| Comm Dropout | 0.1 |

### C.3  FUTURE ARCHITECTURAL CHANGES

We have not conducted systematic optimizations of the proposed architecture. We believe that even principled hyperparameter tuning may significantly improve performance for many of the tasks we have considered in the paper. We briefly mention a few architectural changes which we have studied:

- On the output side, we concatenate the representations of the different RIMs, and use the concatenated representation for learning a policy (in RL experiments) or for predicting the input at the next time step (for bouncing balls as well as all other experiments). We empirically found that adding another layer of (multi-headed) key value attention on the output seems to improve the results. We have not included this change

- In our experiments, we shared the same decoder for all the RIMs, i.e., we concatenate the representations of different RIMS, and feed the concatenated representations to the decoder. In the future it

would be interesting to think of ways to allow a more "structured" decoder. The reason for this is that even if the RIMs generalize to new environments, the shared decoder can fail to do so. So changing the structure of decoder could be helpful.

- For the RL experiments, we also tried providing the previous actions, rewards, language instruction as input to decide the activation of RIMs. This is consistent with the idea of *efference copies* as proposed by von Helmholtz (1867); von Holst & Mittelstaedt (1950), i.e., using copies of motor signals as inputs. Preliminary experiments shows that this improves the performance in Atari games.

## C.4 LANGUAGE MODELING

Table 4: Wikitext-2 results

| Approach | Num. Parameters | Train PPL | Valid PPL | Test PPL |
|----------|-----------------|-----------|-----------|----------|
| LSTM (2-layer) | 21.2M | 39.78 | 109.25 | 102.53 |
| Relational Memory (Santoro et al., 2018) | 11M | n/a | 112.77 | 107.21 |
| **RIMs** (2-layer, $k_T = 6$, $k_A = 6$) | 23.7M | 41.27 | 103.60 | **98.66** |

We investigate the task of word-based language modeling. We ran experiments on the wikitext-2 dataset (Merity et al., 2016). We ran each experiment for a fixed 100 epochs. These results are in Table 4. Our goal in this experiment is to demonstrate the breadth of the approach by showing that RIMs performs well even on datasets which are noisy and drawn from the real-world.

## C.5 COPYING TASK

We used a learning rate of 0.001 with the Adam Optimizer and trained each model for 150 epochs (unless the model was stuck, we found that this was enough to bring the training error close to zero). For the RIMs model we used 600 units split across 6 RIMs (100 units per block). For the LSTM we used a total of 600 units. We did not explore this extensively but we qualitatively found that the results on copying were not very sensitive to the exact number of units.

The sequences to be copied first have 10 random digits (from 0-8), then a span of zeros of some length, followed by a special indicator "9" in the input which instructs the model to begin outputting the copied sequence.

In our experiments, we trained the models with "zero spans" of length 50 and evaluated on the model with "zero spans" of length 200. We note that all the ablations were run with the default parameters (i.e number of keys, values as for RIMs model) for 100 epochs. Tab. 5 shows the effect of two baselines as compared to the RIMs model (a) When we allow the input attention for activation of different RIMs but we dont allow different RIMs to communicate. (b) No Input attention, but we allow different RIMs to communicate with each other. Tab. 5 shows that the proposed method is better than both of these baselines. For copy task, we used 1 head in input attention, and 4 heads for RIMs communication. We note that even with 1 RIM, its not exactly same as a LSTM, because each RIM can still reference itself.

## C.6 SEQUENTIAL MNIST TASK

In this task we considered classifying binary MNIST digits by feeding the pixels to an RNN (in a fixed order scanning over the image). As the focus of this work is on generalization, we introduced a variant on this task where the training digits are at a resolution of 14 x 14 (sequence length of 196). We then evaluated on MNIST digits of different higher resolutions (16 x 16, 19 x 19, and 24 x 24). When re-scaling the images, we used the nearest-neighbor based down-scaling and performed binarization after re-scaling. We trained with a learning rate of 0.0001 and the Adam optimizer. For RIMs we used a total of 600 hidden units split across 6 RIMs (100 units per block). For the LSTM we used a total of 600 units. We ran proposed model as well as baselines for 100 epochs. For sequential MNIST task, we used 1 head in input attention, and 4 heads for RIMs communication.

## C.7 IMITATION LEARNING: ROBUSTNESS TO NOISE IN STATE DISTRIBUTION

Here, we consider imitation learning where we have training trajectories generated from an expert (Table 6). We evaluate our model on continuous control tasks in Mujoco (in our case, Half-Cheetah) (Todorov et al., 2012). We take the rendered images as input and compared the proposed model with recurrent policy (i.e., LSTM). Since, using rendered image of the input does not tell anything about the velocity of the Half-Cheetah, it makes the task partially observable. In order to test how well the proposed model generalizes during test, we add some noise (in the joints of the half-cheetah body). As one can see, after adding noise LSTM baselines performs poorly. On the other hand, for the proposed model, there's also a drop in performance but not as bad as for the LSTM baseline.

Table 5: **Error (CE for last 10 time steps) on the copying task**. Note that while all of the methods are able to learn to copy on the length seen during training, the RIMs model generalizes to sequences longer than those seen during training whereas the LSTM fails catastrophically.

| Approach | Train Length 50 | Test Length 200 |
|---|---|---|
| **RIMs** | **0.00** | **0.00** |
| With input Attention and No Communication | | |
| **RIMs** ($k_T = 4, k_A = 2$) | 2.3 | 1.6 |
| **RIMs** ($k_T = 4, k_A = 3$) | 1.7 | 4.3 |
| **RIMs** ($k_T = 5, k_A = 2$) | 2.5 | 4,7 |
| **RIMs** ($k_T = 5, k_A = 3$) | 0.4 | 4.0 |
| **RIMs** ($k_T = 5, k_A = 4$) | 0.2 | 0.7 |
| **RIMs** ($k_T = 6, k_A = 2$) | 3.3 | 2.4 |
| **RIMs** ($k_T = 6, k_A = 3$) | 1.2 | 1.0 |
| **RIMs** ($k_T = 6, k_A = 4$) | 0.7 | 5.0 |
| **RIMs** ($k_T = 6, k_A = 5$) | 0.22 | 0.56 |
| With No input Attention and Full Communication | | |
| **RIMs** ($k_T = 6, k_A = 6, h_{dim} = 600$) | 0.0 | 0.7 |
| **RIMs** ($k_T = 5, k_A = 5, h_{dim} = 500$) | 0.0 | 1.7 |
| **RIMs** ($k_T = 2, k_A = 2, h_{dim} = 256$) | 0.0 | 2.9 |
| **RIMs** ($k_T = 2, k_A = 2, h_{dim} = 512$) | 0.0 | 1.8 |
| **RIMs** ($k_T = 1, k_A = 1, h_{dim} = 512$) | 0.0 | 0.2 |

Table 6: **Imitation Learning:** Results on the half-cheetah imitation learning task. RIMs outperforms a baseline LSTM when we evaluate with perturbations not observed during training (left). An example of an input image fed to the model (right).

| Method / Setting | Training Observed Reward | Perturbed States Observed Reward | |
|---|---|---|---|
| LSTM (Recurrent Policy) | $5400 \pm 100$ | $2500 \pm 300$ | 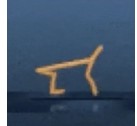 |
| **RIMs** ($k_T = 6, k_A = 3$) | $5300 \pm 200$ | $3800 \pm 200$ | |
| **RIMs** ($k_T = 6, k_A = 6$) | $5500 \pm 100$ | $2700 \pm 400$ | |
| **RIMs** (without Input attention) | $5400 \pm 100$ | $3200 \pm 50$ | |

We use the convolutional network from (Ha & Schmidhuber, 2018) as our encoder, a GRU (Chung et al., 2015) with 600 units as deterministic path in the dynamics model, and implement all other functions as two fully connected layers of size 256 with ReLU activations. Since, here we are using images as input, which makes the task, partially observable. Hence, we concatenate the past 4 observations, and then feed the concatenated observations input to GRU (or our model). For our model, we use 6 RIMs, each of size 100, and we set $k_a = 3$. We follow the same setting as in (Hafner et al., 2018; Sodhani et al., 2019). We also compare the proposed method to the baseline where we dont include input attention (or top-down attention). AS 6 shows, there's a decline in performance if we dont use input attention, hence justifying the importance

## C.8  GENERALIZATION TO DISTRACTORS: ALGORITHM IMPLEMENTATION DETAILS

We evaluate the proposed framework using Adavantage Actor-Critic (A2C) to learn a policy $\pi_\theta(a|s, g)$ conditioned on the goal. To evaluate the performance of proposed method, we use a range of maze multi-room tasks from the gym-minigrid framework (Chevalier-Boisvert & Willems, 2018) and the A2C implementation from (Chevalier-Boisvert & Willems, 2018). For the maze tasks, we used agent's relative distance to the absolute goal position as "goal".

For the maze environments, we use A2C with 48 parallel workers. Our actor network and critic networks consist of two and three fully connected layers respectively, each of which have 128 hidden units. The encoder network is also parameterized as a neural network, which consists of 1 fully connected layer. We use RMSProp with an initial learning rate of 0.0007 to train the models. Due to the partially observable nature of the environment, we further use a LSTM to encode the state and summarize the past observations.

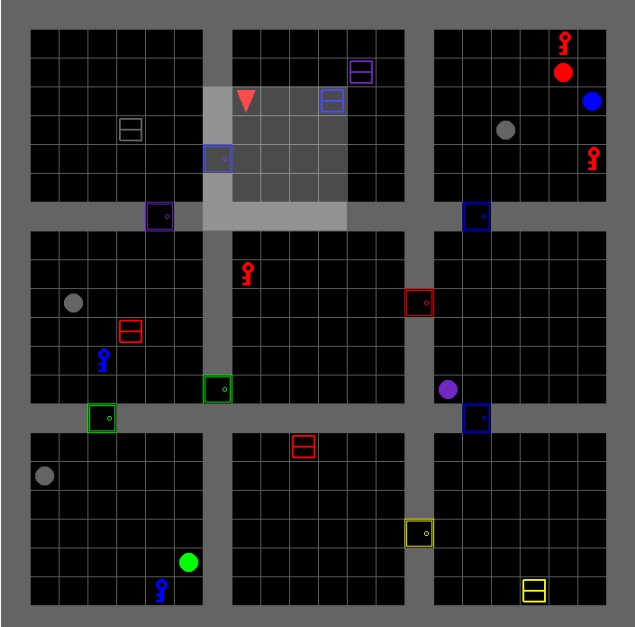

Figure 7: An example of the minigrid task.

## C.9 MINIGRID ENVIRONMENTS FOR OPENAI GYM

The MultiRoom environments used for this research are part of MiniGrid, which is an open source gridworld package[2]. This package includes a family of reinforcement learning environments compatible with the OpenAI Gym framework. Many of these environments are parameterizable so that the difficulty of tasks can be adjusted (e.g., the size of rooms is often adjustable).

### C.9.1 THE WORLD

In MiniGrid, the world is a grid of size NxN. Each tile in the grid contains exactly zero or one object. The possible object types are wall, door, key, ball, box and goal. Each object has an associated discrete color, which can be one of red, green, blue, purple, yellow and grey. By default, walls are always grey and goal squares are always green.

### C.9.2 REWARD FUNCTION

Rewards are sparse for all MiniGrid environments. In the MultiRoom environment, episodes are terminated with a positive reward when the agent reaches the green goal square. Otherwise, episodes are terminated with zero reward when a time step limit is reached. In the FindObj environment, the agent receives a positive reward if it reaches the object to be found, otherwise zero reward if the time step limit is reached.

The formula for calculating positive sparse rewards is $1 - 0.9 * (step\_count/max\_steps)$. That is, rewards are always between zero and one, and the quicker the agent can successfully complete an episode, the closer to $1$ the reward will be. The $max\_steps$ parameter is different for each environment, and varies depending on the size of each environment, with larger environments having a higher time step limit.

### C.9.3 ACTION SPACE

There are seven actions in MiniGrid: turn left, turn right, move forward, pick up an object, drop an object, toggle and done. For the purpose of this paper, the pick up, drop and done actions are irrelevant. The agent can use the turn left and turn right action to rotate and face one of 4 possible directions (north, south, east, west). The move forward action makes the agent move from its current tile onto the tile in the direction it is currently facing, provided there is nothing on that tile, or that the tile contains an open door. The agent can open doors if they are right in front of it by using the toggle action.

---

[2]https://github.com/maximecb/gym-minigrid

### C.9.4 OBSERVATION SPACE

Observations in MiniGrid are partial and egocentric. By default, the agent sees a square of 7x7 tiles in the direction it is facing. These include the tile the agent is standing on. The agent cannot see through walls or closed doors. The observations are provided as a tensor of shape 7x7x3. However, note that these are not RGB images. Each tile is encoded using 3 integer values: one describing the type of object contained in the cell, one describing its color, and a flag indicating whether doors are open or closed. This compact encoding was chosen for space efficiency and to enable faster training. The fully observable RGB image view of the environments shown in this paper is provided for human viewing.

### C.9.5 LEVEL GENERATION

The level generation in this task works as follows: (1) Generate the layout of the map (X number of rooms with different sizes (at most size Y) and green goal) (2) Add the agent to the map at a random location in the first room. (3) Add the goal at a random location in the last room. A neural network parameterized as CNN is used to process the visual observation.

We follow the same architecture as (Chevalier-Boisvert & Willems, 2018) but we replace the LSTM layer with BlockLSTM.

### C.10 BOUNCING BALLS

We use the bouncing-ball dataset from (Van Steenkiste et al., 2018). The dataset consists of 50,000 training examples and 10,000 test examples showing ∼50 frames of either 4 solid balls bouncing in a confined square geometry, 6-8 balls bouncing in a confined geometry, or 3 balls bouncing in a confined geometry with a random occluded region. In all cases, the balls bounce off the wall as well as off one another. We train baselines as well as proposed model for about 100 epochs using 0.0007 as learning rate and using Adam as optimizer (Kingma & Ba, 2014). We use the same architecture for encoder as well as decoder as in (Van Steenkiste et al., 2018). We train the proposed model as well as the baselines for 100 epochs. Our goal in this section is to give more thorough experimental results omitted from the main paper for the sake of brevity. Below, we highlight a few different results.

### C.10.1 DIFFERENT RIMS ATTEND TO DIFFERENT BALLS

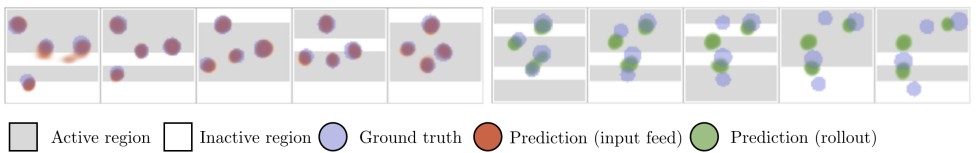

Figure 8: **Different RIMs attending to Different Balls**. For understanding what each RIM is actually doing, we associate each with a separate encoder, which are spatially masked. Only 4 encoders can be active at any particular instant and there are four different balls. We did this to check if there would be the expected geometric activation of RIMs. 1.) Early in training, RIM activations correlated more strongly with the locations of the four different balls. Later in training, this correlation decreased and the active strips did not correlate as strongly with the location of balls. As the model got better at predicting the location, it needed to attend less to the actual objects. The top row shows every 5th frame when the truth is fed in and the bottom shows the results during rollout. The gray region shows the active block. In the top row, the orange corresponds to the prediction and in the bottom, green corresponds to the prediction.

In order to visualize what each RIM is doing, we associate each RIM with a different encoder. By performing spatial masking on the input, we can control the possible spatial input to each RIM. We use six non-overlapping horizontal strips and allow only 4 RIMs to be active at a time (shown in Fig. 8). The mask is fixed mask of zeros with a band of ones that is multiplied by the input to each encoder. Therefore, each of the 6 encoders gets 1/6th of the input. The goal was to see how the RIM activation patterns changed/correlated with the locations of the balls. We find that early in training, the RIMs' activations are strongly correlated with the location of the 4 balls. However, after training has proceeded for some time this correlation deteriorates. This is likely because the predictable dynamics of the system do not necessitate constant attention.

### C.10.2 COMPARISON WITH LSTM BASELINES

In Figures 9, 10, 11, and 12 we highlight different baselines and how these compare to the proposed RIMs model.

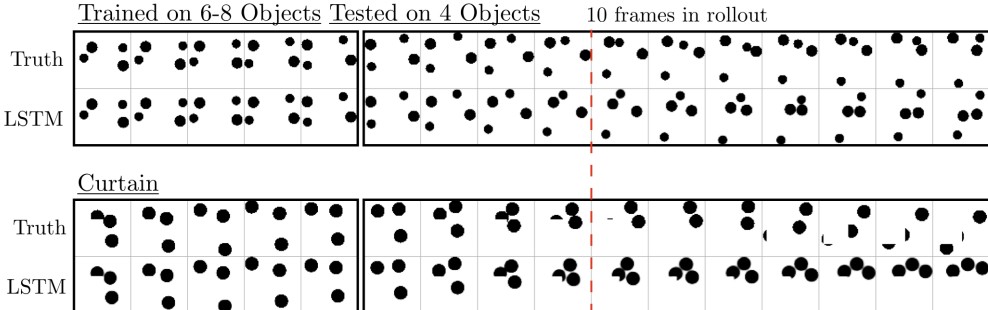

Figure 9: **Example of the other LSTM baselines**. For the 2 other experiments that we consider, here we show example outputs of our LSTM baselines. In each row, the top panel represents the ground truth and the bottom represents the prediction. All shown examples use an LSTM with 250 hidden units, as shown in Fig. 4. Frames are plotted every 3rd time step. The red line marks 10 rollout frames. This is marked because after this we do not find BCE to be a reliable measure of dissimilarity.

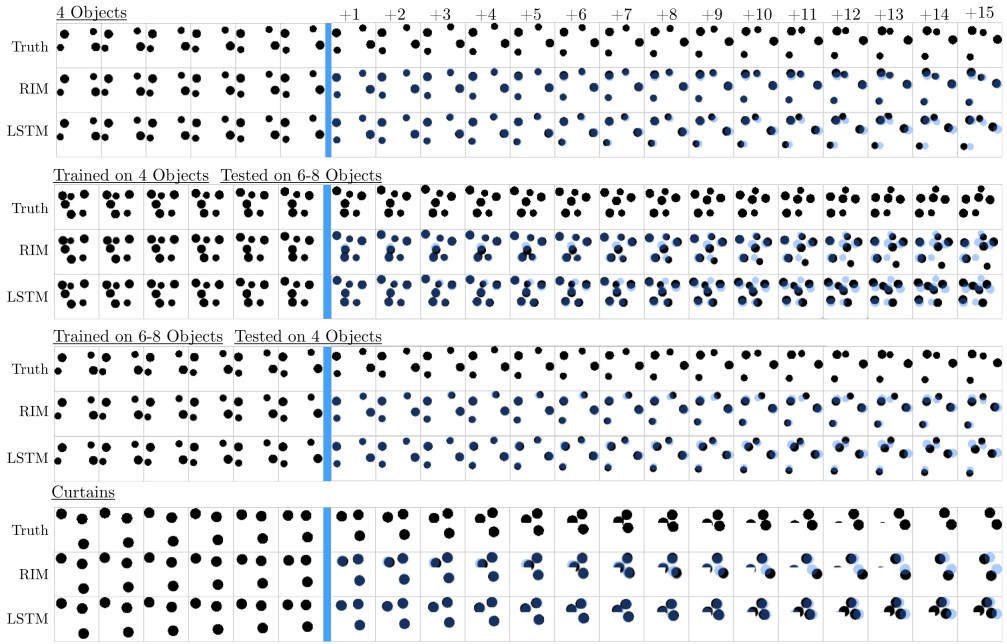

Figure 10: **Comparison of RIMs to LSTM baseline**. For 4 different experiments in the text, we compare RIMs to two different LSTM baselines. In all cases we find that during rollout, RIMs perform better than the LSTMs at accurately capturing the trajectories of the balls through time. Due to the number of hard collisions, accurate modeling is very difficult. In all cases, the first 15 frames of ground truth are fed in (last 6 shown) and then the system is rolled out for the next 15 time steps, computing the binary cross entropy between the prediction and the true balls at each instant, as in Van Steenkiste et al. (2018). See the Appendix for losses over the entire 35 frame rollout trajectory. In the predictions, the transparent blue shows the ground truth, overlaid to help guide the eye.

### C.10.3 OCCLUSION

In Fig. 13, we show the performance of RIMs on the curtain dataset. We find RIMs are able to track balls through the occlusion without difficulty. Note that the LSTM baseline, is also able to track the ball through the "invisible" curtain.

### C.10.4 STUDY OF TRANSFER

It is interesting to ask how models trained on a dataset with 6-8 balls perform on a dataset with 4 balls. In Fig. 14 we show predictions during feed-in and rollout phases.

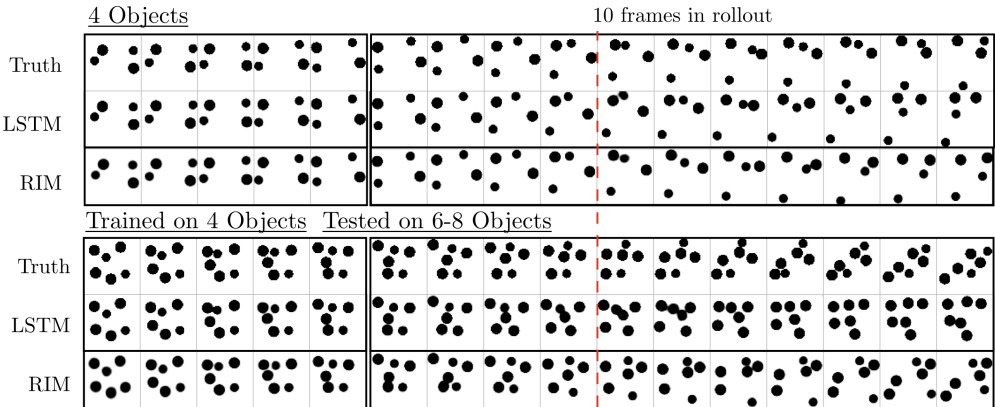

Figure 11: **Comparison between RIMs and LSTM baseline**. For the 4 ball task and the 6-8 ball extrapolation task, here we show an example output of from our LSTM baseline and from RIMs. All shown examples use an LSTM with 250 hidden units, as shown in Fig. 4. Frames are plotted every 3rd time step. The red line marks 10 rollout frames. This is marked because after this we do not find BCE to be a reliable measure of dissimilarity.

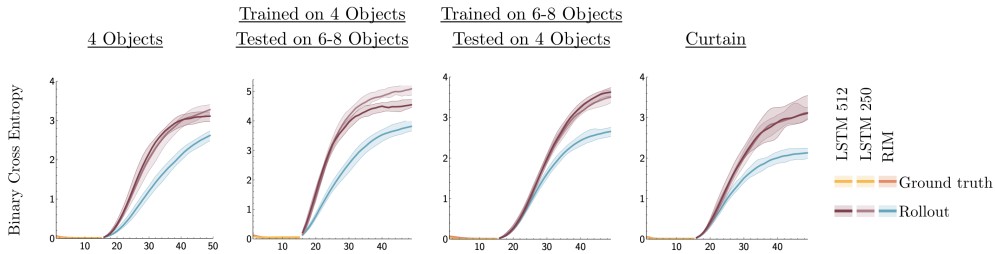

Figure 12: **Comparison of RIMs to LSTM baseline**. For 4 different experiments in the text, we compare RIMs to two different LSTM baselines. In all cases we find that during rollout, RIMs perform better than the LSTMs at accurately capturing the trajectories of the balls through time. Due to the number of hard collisions, accurate modeling is very difficult.

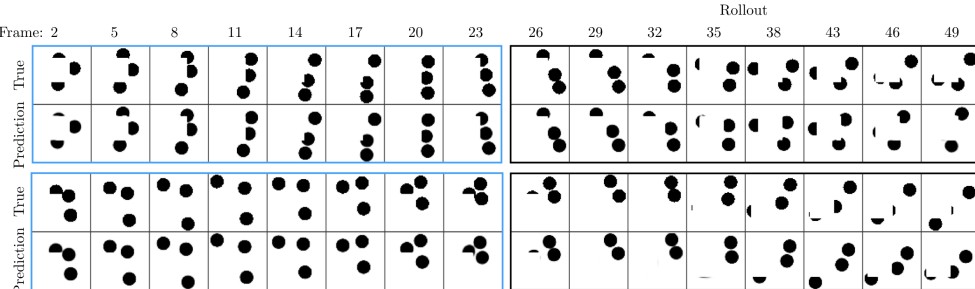

Figure 13: **RIMs on dataset with an occlusion**. We show two trajectories (top and bottom) of three balls. For the left frames, at each step the true frame is used as input. On the right, outlined in black, the previous output is used as input.

## C.11 ABLATIONS

We present one ablation in addition to the ones in Section 4.4. In this experiment, we study the effect on input attention (i.e top down attention) as well as the use of multi-headed head key-value attention. We compare the proposed model (with input attention as well as multi-headed key value attention) with 2 baselines: (a) In which we remove the input attention (and force all the RIMs to communicate with each other) (b) We use 1 head for key value attention as compared to multi-headed key-value attention. Results comparing the proposed model, with these two baselines is shown in Fig. 15.

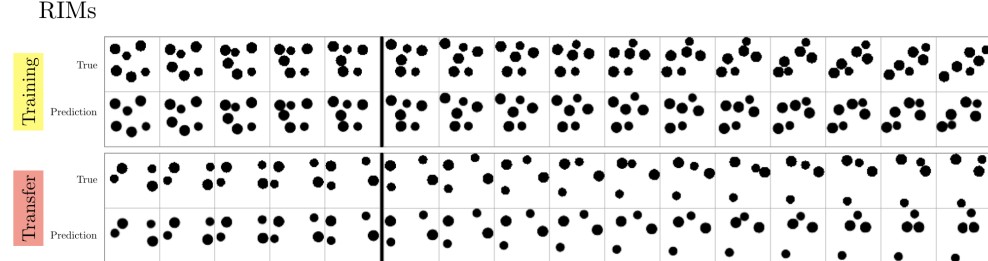

Figure 14: **RIMs transferred on new data**. We train the RIMs model on the 6-8 ball dataset (as shown in the top row). Then, we apply the model to the 4 ball dataset, as shown in the bottom.

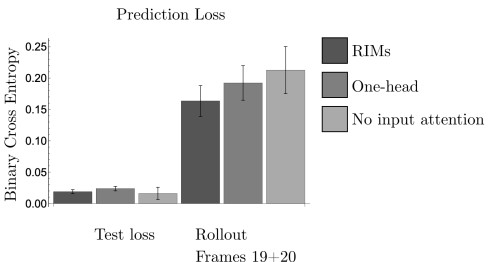

Figure 15: **Ablation loss** For the normal, a one-head model, and without input attention, we show the loss during training and the loss for the 4th and 5th frame of rollout. We find that the one-head and without input attention models perform worse than the normal RIMs model during the rollout phase.

In Fig. 16, we show the predictions that result from the model with only one active head.

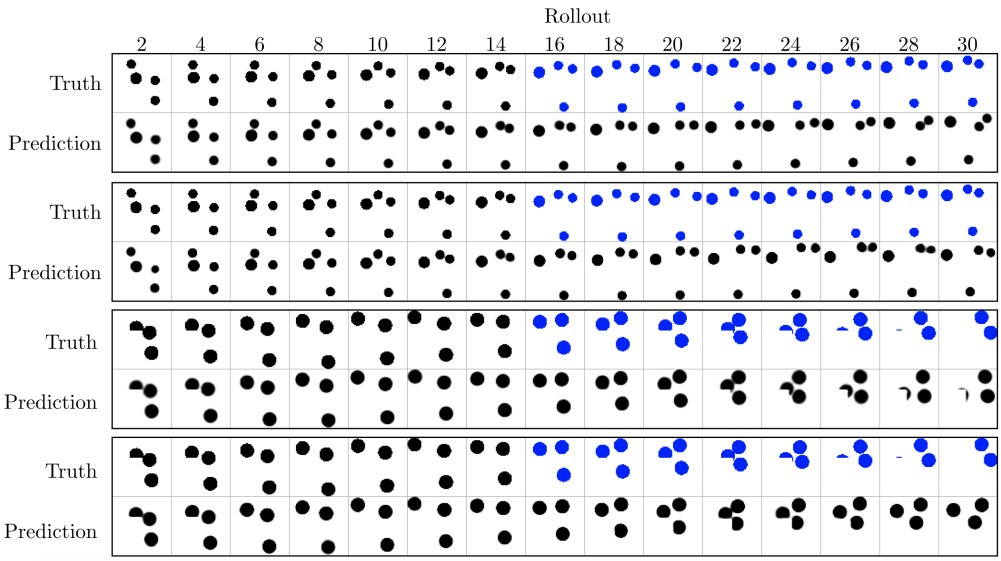

Figure 16: **One head and no attention** Using one head and no attention models, we show the rollout predictions in blue. On top we show results on the 4 ball dataset and on the bottom we show results on the curtains dataset.

| Environment | LSTM-PPO | RIMs-PPO |
|---|---|---|
| Alien | 1612 ± 44 | **2152** ± 81 |
| Amidar | 1000 ± 58 | **1800** ± 43 |
| Assault | 4000 ± 213 | **5400** ± 312 |
| Asterix | 3090 ± 420 | **21040** ± 548 |
| Asteroids | 1611.0 ± 200 | **3801** ± 89 |
| Atlantis | 3280000 ± 200000 | **3500000** ± 120000 |
| BankHeist | 1153 ± 23 | **1195** ± 4 |
| BattleZone | 21000 ± 232.0 | **22000** ± 324 |
| BeamRider | 698 ± 100 | **5320** ± 300 |
| Bowling | 30 ± 5 | 42 ± 13 |
| Boxing | 80 ± 3 | **95** ± 10 |
| Breakout | 593 ± 90 | 590 ± 10 |
| Centipede | 4600 ± 312 | **5534** ± 283 |
| ChopperCommand | 11000 ± 790 | **12303** ± 412 |
| CrazyClimber | **138000** ± 2412 | 132039 ± 1221 |
| DemonAttack | 26320 ± 3234 | **230324** ± 4032 |
| DoubleDunk | **-3.0** ± 0.5 | -3.8 ± 0.3 |
| Enduro | 1600 ± 200 | **2800** ± 232 |
| FishingDerby | 20 ± 4 | **38** ± 8 |
| Freeway | 29 ± 2 | **33** ± 2 |
| Gopher | 7000.0 ± 402 | **33000** ± 2210 |
| Gravitar | 500 ± 100 | **1090** ± 80 |
| IceHockey | -5 ± 0.3 | -4 ± 1 |
| Jamesbond | 425 ± 25 | **800** ± 100 |
| Kangaroo | **13000** ± 500 | 1800 ± 400 |
| Krull | **10000** ± 500 | 7900 ± 200 |
| KungFuMaster | 28000 ± 2000 | **51000** ± 800 |
| NameThisGame | 4200 ± 400 | **6800** ± 300 |
| Pong | 20 ± 1 | 20 ± 1 |
| PrivateEye | 90 ± 3 | **100** ± 0 |
| Qbert | 22000 ± 300 | **22500** ± 400 |
| Riverraid | 7500 ± 300 | **12000** ± 100 |
| RoadRunner | 53000 ± 120 | **53430** ± 300 |
| Robotank | 3 ± 1 | **11** ± 2 |
| SpaceInvaders | 1600 ± 40 | **2800** ± 80 |
| StarGunner | 35000 ± 800 | **70000** ± 1200 |
| TimePilot | 4000 ± 100 | **10000** ± 689 |
| UpNDown | 70000 ± 6000 | **390000** ± 20000 |
| VideoPinball | 90000 ± 5000 | **220000** ± 9000 |
| WizardOfWor | 3833 ± 400 | **10800** ± 700 |
| Zaxxon | 200 ± 100 | **15000** ± 600 |

Table 7: Scores obtained using PPO with the LSTM architecture and PPO with the RIMs architecture with $k_A = 5$.

## C.12   ATARI

We used open-source implementation of PPO from (Kostrikov, 2018) with default parameters. We ran the proposed algorihtm with 6 RIMs, and kept the number of activated RIMs to 4/5. We have not done any hyper-parameter search for Atari experiments.

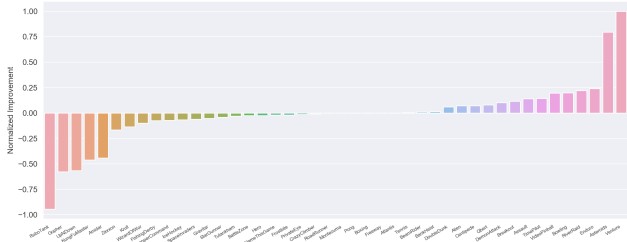

Figure 17: A comparison showing relative improvement of RIMs with $k_A = 5$ over a $k_A = 4$ baseline. Using $k_A = 5$ performs slightly worse than $k_A = 4$ but still outperforms PPO, and has similar results across the majority of games.

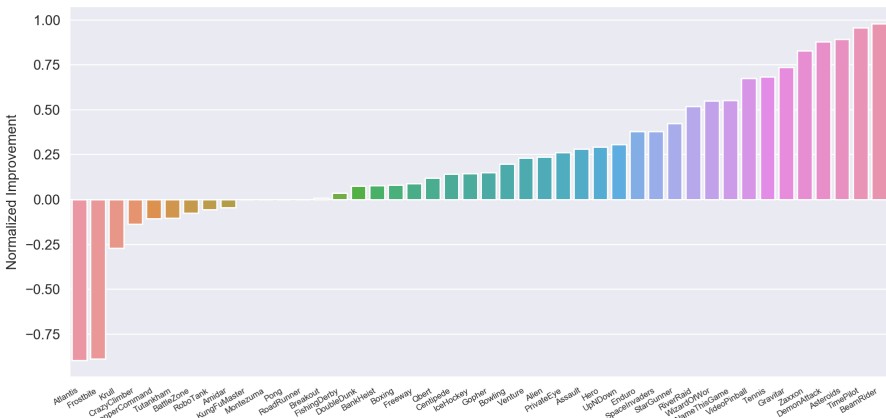

Figure 18: RIMs-PPO relative score improvement over LSTM-PPO baseline (Schulman et al., 2017) across all Atari games averaged over 3 trials per game. In both cases PPO was used with the exact same settings with the only change being the choice of the recurrent architecture (RIMs with $k_A = 5$).

### C.12.1 TRANSFER ON ATARI

As a very preliminary result, we investigate feature transfer between randomly selected Atari games. In order to study this question, we follow the experimental protocol of Rusu et al. (2016).

We start by training RIMs on three source games (Pong, River Raid, and Seaquest) and test if the learned features transfer to a different subset of randomly selected target games (Alien, Asterix, Boxing, Centipede, Gopher, Hero, James Bond, Krull, Robotank, Road Runner, Star Gunner, and Wizard of Wor). We observe, that RIMs result in positive transfer in 9 out of 12 target games, with three cases of negative transfer. On the other hand progressive networks (Rusu et al., 2016) result in positive transfer in 8 out of 12 target games, and two cases of negative transfer. We also compare to LSTM baseline, which yields positive transfer in 3 of 12 games.

### C.13 BOUNCING MNIST: DROPPING OFF RIMS

We use the Stochastic Moving MNIST (SM-MNIST) (Denton & Fergus, 2018) dataset which consists of sequences of frames of size $64 \times 64$, containing one or two MNIST digits moving and bouncing off the walls. Training sequences are generated on the fly by sampling two different MNIST digits from the training set (60k total digits) and two distinct trajectories.

Here, we show the effect of masking out a particular RIM and study the effect of the masking on the ensemble of RIMs. Ideally, we would want different RIMs not to co-adapt with each other. So, masking out a particular RIM should not really effect the dynamics of the entire model. We show qualitative comparisons in Fig. 19, 20, 21, 22, 23. In each of these figures, the model gets the ground truth image as input for first 5 time steps, and then asked to simulate the dynamics for next 25 time-steps. We find that sparsity is needed otherwise different RIMs co-adapt with each other (for ex. see Fig. 20, 22, 23). We tried similar masking experiments for different

models like RMC, Transformers, EntNet (which learns a mixture of experts), LSTMs, but all of them failed to do anything meaningful (after masking). We suspect this is partly due to learning a homogeneous network.

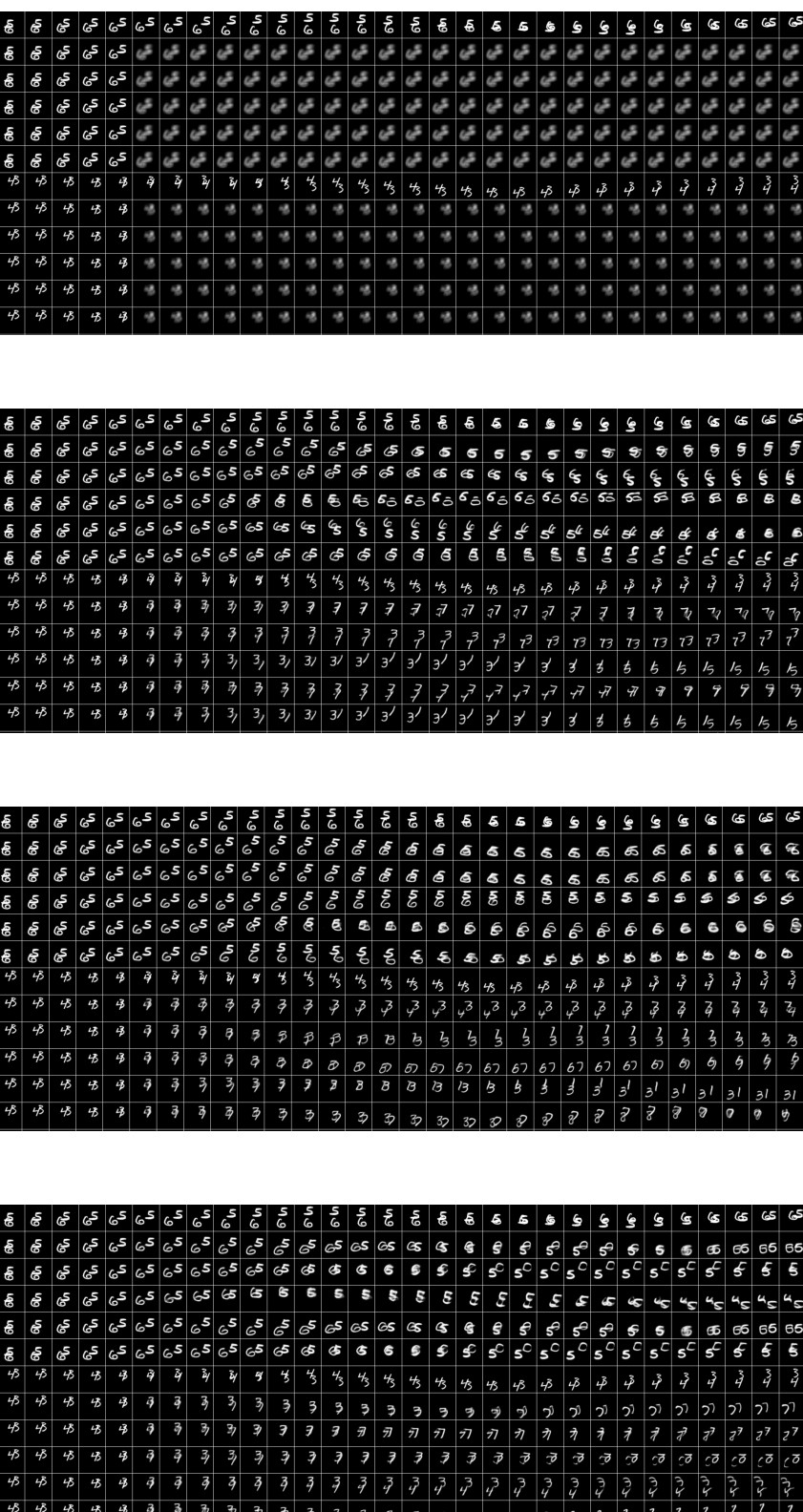

Figure 19: 4 RIMs, (top k = 2). Each sub-figure shows the effect of masking a particular RIM and studying the effect of masking on the other RIMs. For example, the top figure shows the effect of masking the first RIM, the second figure shows the effect of masking the second RIM etc.

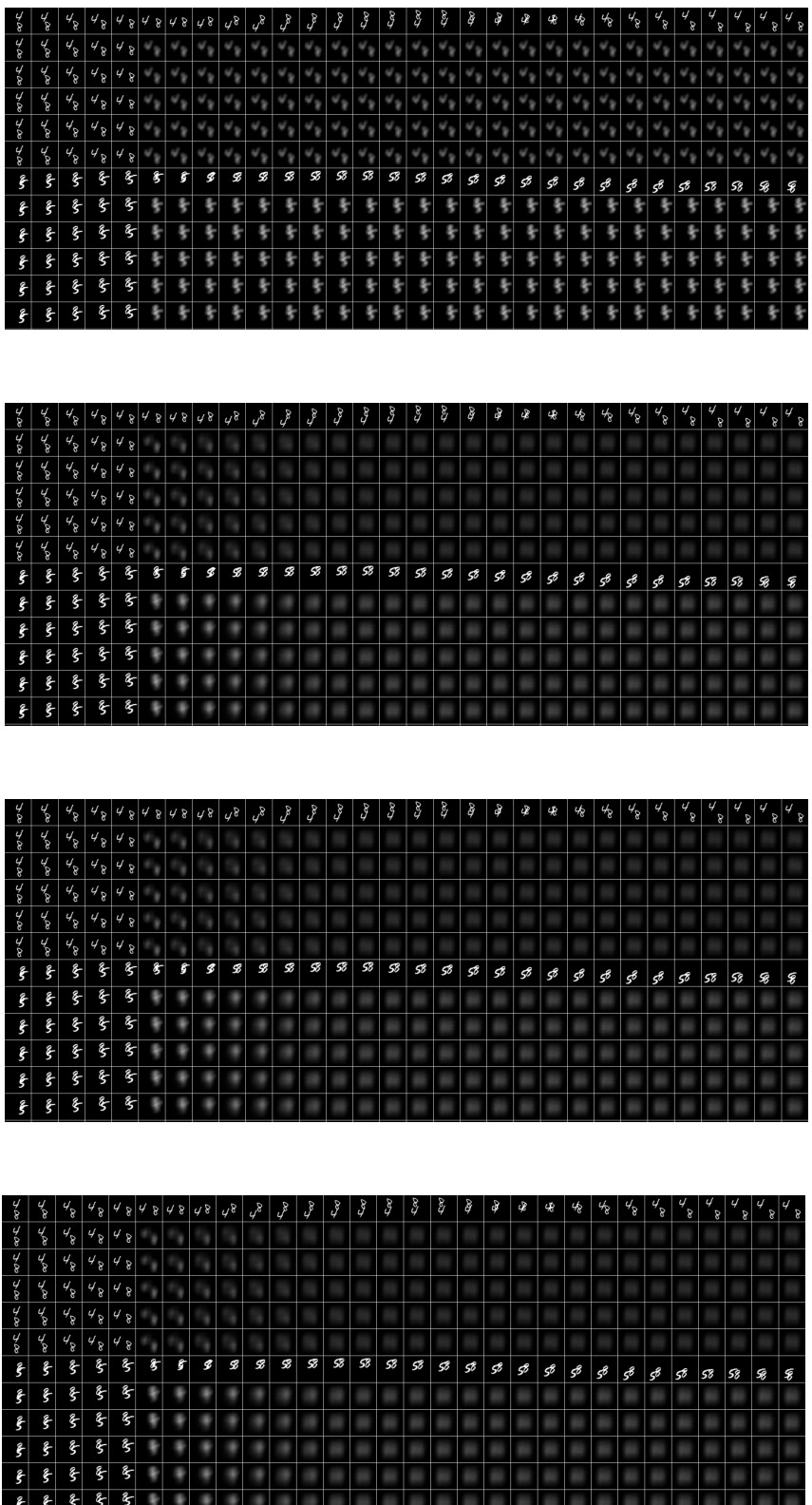

Figure 20: 4 RIMs, (top k = 3). Each sub-figure shows the effect of masking a particular RIM and studying the effect of masking on the other RIMs. For example, the top figure shows the effect of masking the first RIM, the second figure shows the effect of masking the second RIM etc.

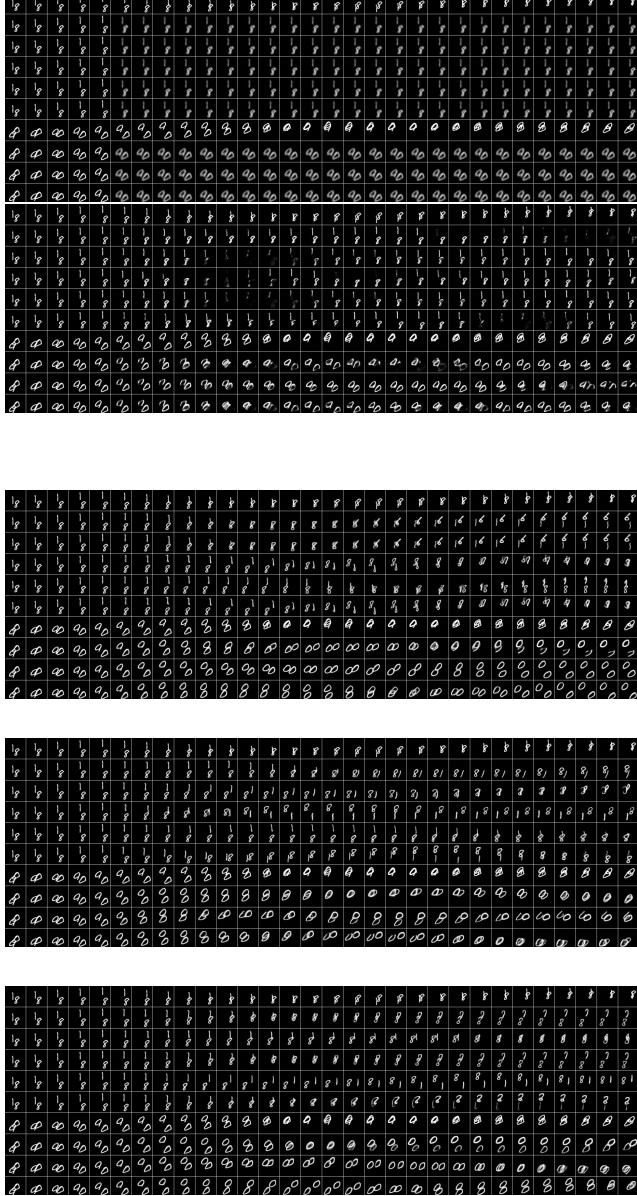

Figure 21: 400dim, 5 RIMs, (top k = 2). Each sub-figure shows the effect of masking a particular RIM and studying the effect of masking on the other RIMs. For example, the top figure shows the effect of masking the first RIM, the second figure shows the effect of masking the second RIM etc.

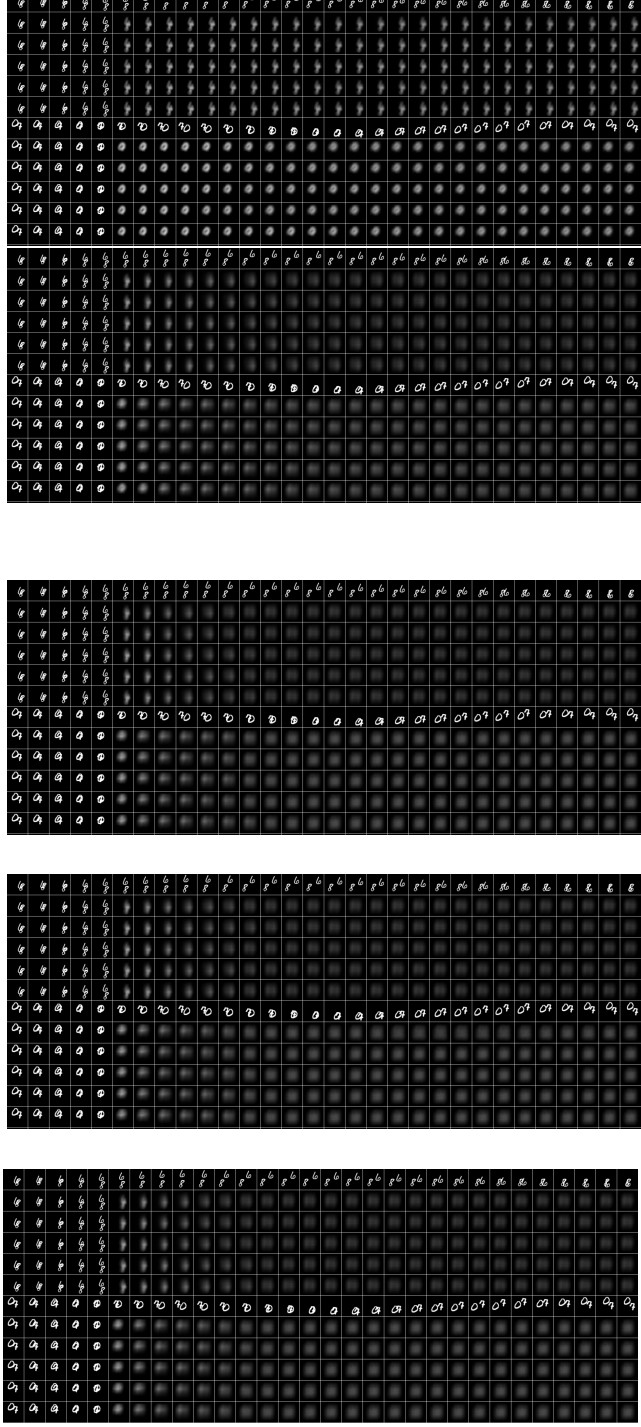

Figure 22: 400dim, 5 blocks, (top k = 3). Each sub-figure shows the effect of masking a particular RIM and studying the effect of masking on the other RIMs. For examples, the top figure shows the effect of masking the first RIM, the second figure shows the effect of masking the second RIM etc.

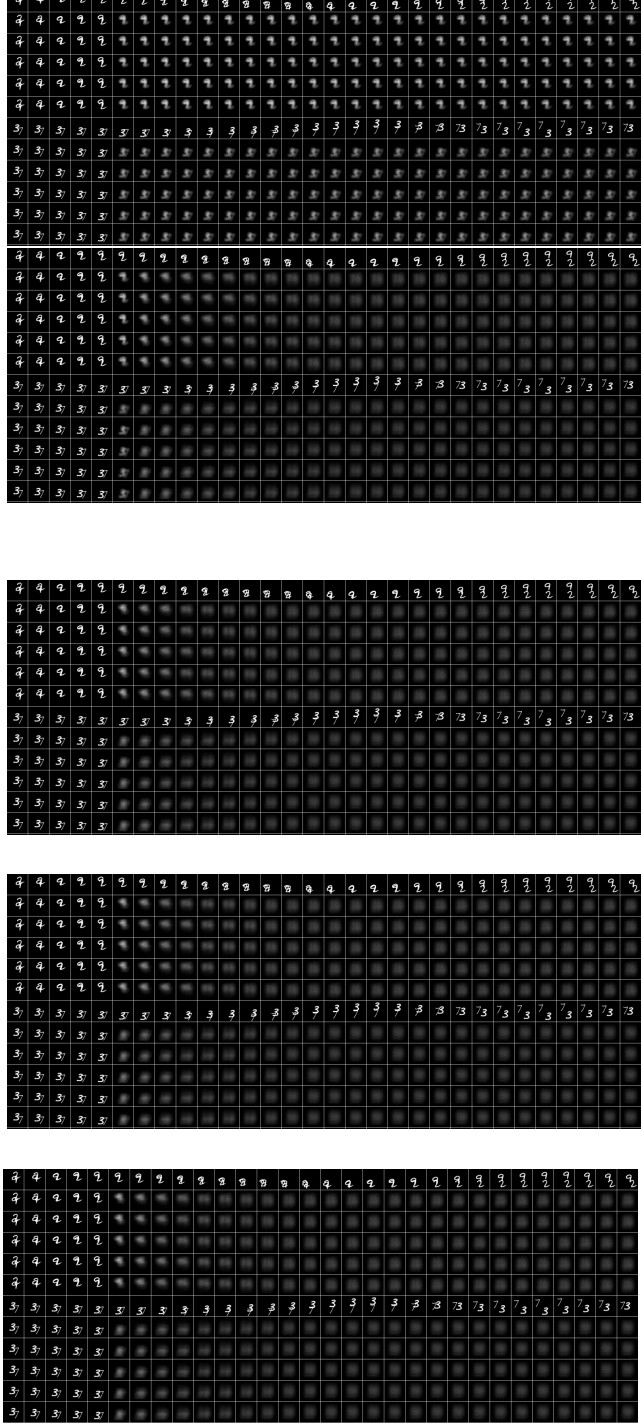

Figure 23: 400dim, 5 blocks, (top k = 4). Each sub-figure shows the effect of masking a particular RIM and studying the effect of masking on the other RIMs. For example, the top figure shows the effect of masking the first RIM, the second figure shows the effect of masking the second RIM etc.

### C.13.1   ATARI RESULTS: COMPARISON WITH LSTM-PPO

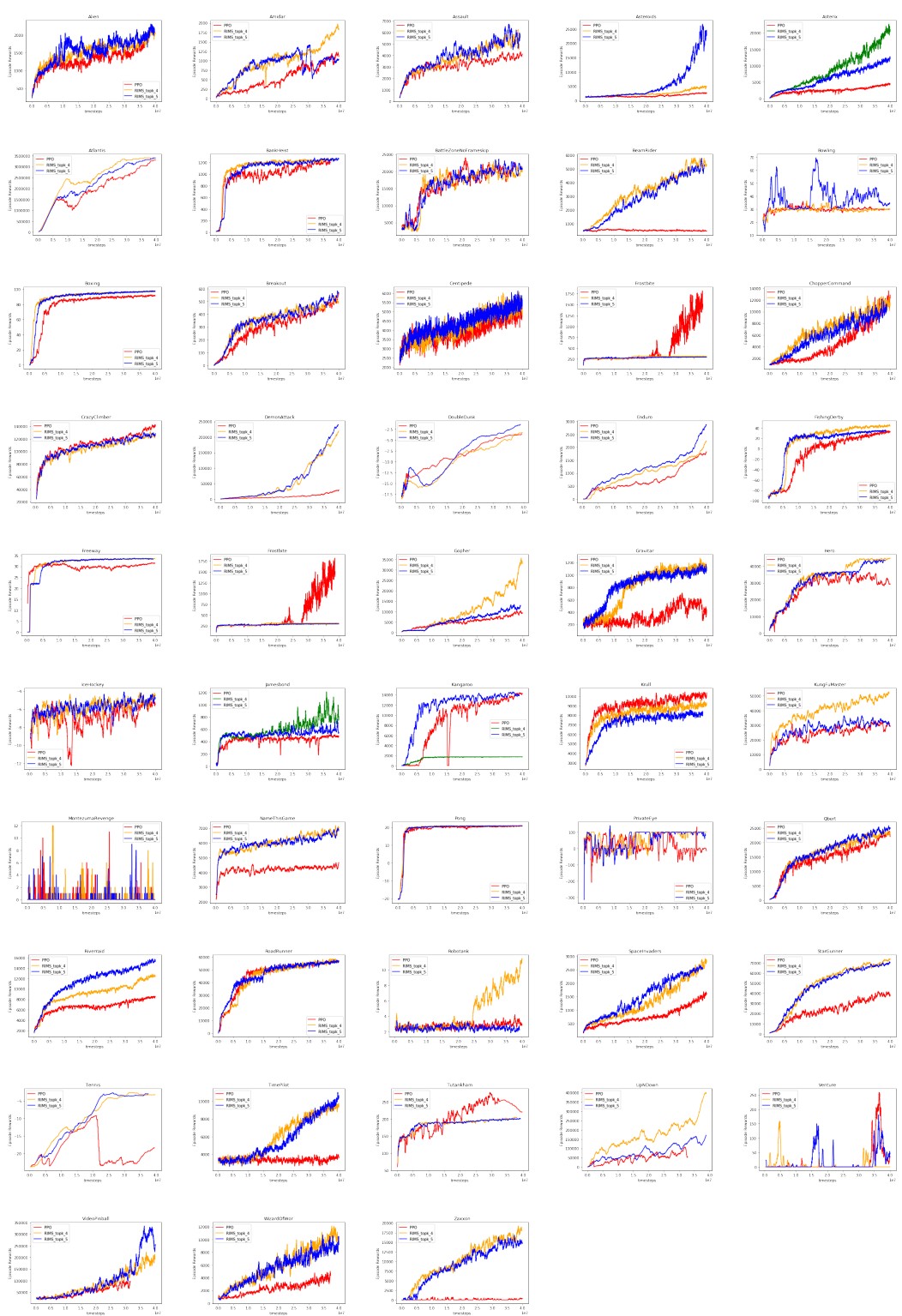

Figure 24: **Comparing RIMs-PPO with LSTM-PPO:** Learning curves for $k_A = 4$, $k_A = 5$ RIMs-PPO models and the LSTM-PPO baseline across all Atari games.

### C.13.2 ATARI RESULTS: NO INPUT ATTENTION

Here we compare the proposed method to the baseline, where we dont use input attention, and we force different RIMs to communicate with each at all the time steps.

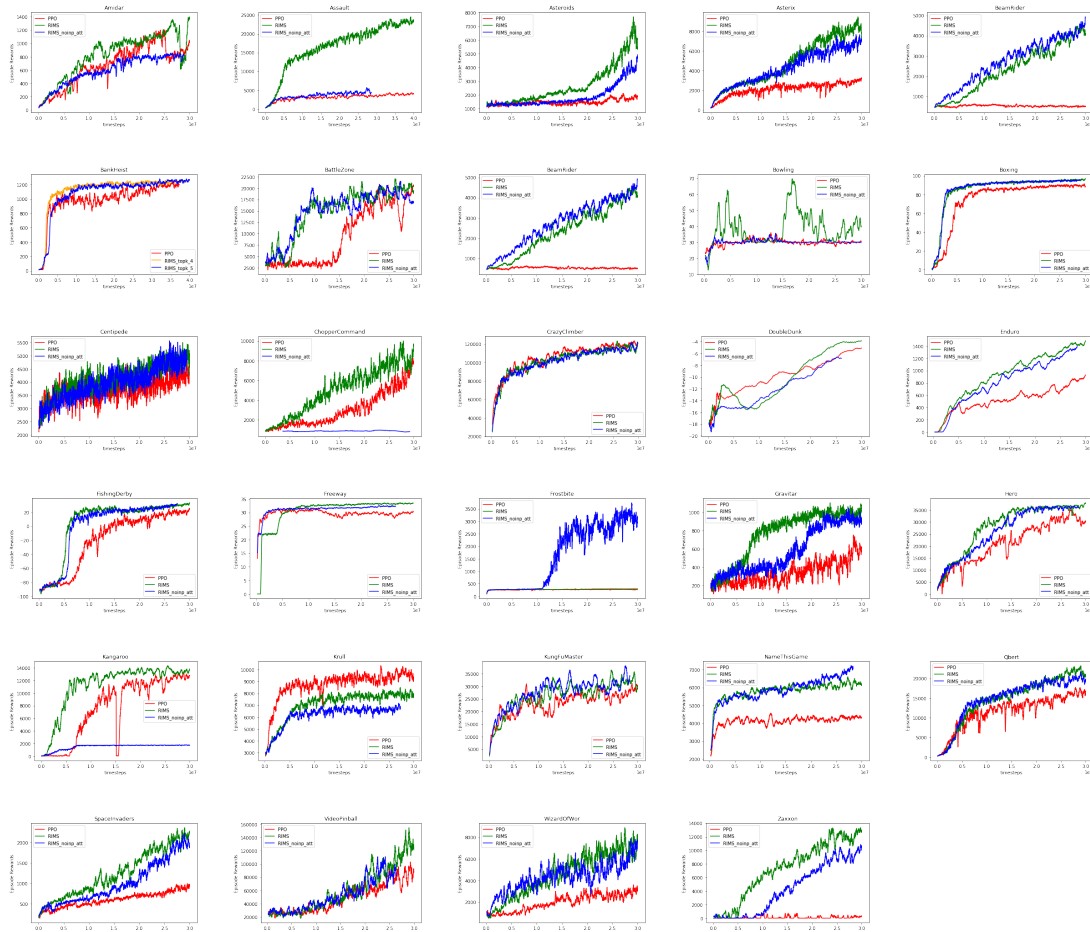

Figure 25: **Baseline agent with no input attention mechanism:** Here we compare the RIMs to the baseline, where their is no input attention (i.e., top down attention) as well as all the RIMs communicate with each other at all the time steps. Learning curves for RIMs-PPO models, Baseline Agent, the LSTM-PPO baseline across 30 Atari games.

