# OpenReview forum: "Recurrent Independent Mechanisms"
_ICLR.cc/2020/Conference — Reject_

### Official Review · AnonReviewer1 · 2019-10-17
**Official Blind Review #1**

**Rating:** 6

**Review:**

This work seems to propose an alternative to general RNN so that  the dynamics of sequential data can be better captured. The work is based on a hypothesis that a causal process can be modeled by "independent" modules and sparse interactions.

The paper is written in very fluent English, but the style is less technical. The impression is that the philosophical arguments and the machine learning realization have a gap in between. There is no much rigorous mathematical definition or derivation to back up the entire development. The mathematical symbols are a bit loosely defined. For example, it is unclear to the reviewer if h is a scalar or a vector.

The RIM idea seems to be derived from some ideas from the ``'causality community' . But the authors did not elaborate how significant will this structure change the state of art. In particular, reading the abstract or the introduction does not shed much light on what are the challenges of now the ML community is facing, and how this proposed RIM idea is going to help. This may have been obvious to the authors, but spelling them out may help the reviewer/readers to understand the contribution. The related work section helped a bit, but still unclear.

The reviewer feels that the paper stands at a high level in general, but lacks concrete examples/applications for general readers to appreciate the significance. Perhaps trying to re-organize this part could greatly help the readability.

The mathematical descriptions in 2.2, 2.3 and 2.4 are very hard to follow. There is no cost function (for training) around, but there are discussions of  'gradient'. Gradient of which function?

**Experience Assessment:**

I do not know much about this area.

**Review Assessment: Checking Correctness Of Derivations And Theory:**

N/A

**Review Assessment: Checking Correctness Of Experiments:**

I did not assess the experiments.

**Review Assessment: Thoroughness In Paper Reading:**

I made a quick assessment of this paper.

---

> ### Author Response · Authors · 2019-11-07
> **“Independent Mechanisms and RIMs” (1/2)**
>
> Thank you for the feedback.  We appreciate your comment that the “paper stands at a high level in general”.
>
>
> “Independent Mechanisms and RIMs”
>
> In the causality literature, it is a common assumption to view any real-world distribution as a product of causal mechanisms (i.e., causal conditionals). A change in such a distribution (e.g., when moving from one setting/domain to a related one) will always be due to changes in at least one of those mechanisms. Consistent with the independence principle, we hypothesize that such changes tend to manifest themselves in a sparse or local way, i.e., they should usually not affect all factors simultaneously. In contrast, if we consider a non-causal factorization, then many terms will be affected simultaneously as we change one of the physical mechanisms responsible for a system’s statistical dependencies. Such a factorization may be described as entangled, a term that has recently gained popularity in machine learning (Bengio et al., 2012; Locatello et al., 2018; Suter et al., 2018). The notion of invariant, autonomous, and independent mechanisms has appeared in various guises throughout the history of causality research.
> Our high-level argument is that we want to create a recurrent architecture which makes it easy for the model to capture independent mechanisms.  Thus we view RIMs as a recurrent architecture which is motivated by the notion of causal independence.  How this relates to other research in the space of recurrent architectures is discussed in related work and we’ve found that none have the same kind of strong modularity captured by RIMs, although some of them share certain specific aspects (for example neural turing machines and relational memory cores both benefit from having multiple separate memory cells).
>
> We think that at least in the case of monolithic architectures (like normal LSTMs), there is an intuitive argument for why very difficult for them to learn fully independent mechanisms.  The simple reason for this is that fully-connected layers are used over the entire hidden state (per each time step) so to keep information perfectly-separated between the hidden states, the majority of the parameters need to be zero.  In practice, the optimization has no incentive to learn this modularity perfectly, but would instead settle for only having it hold well enough to fit the training data, which in turn leads to poor generalization to changing environments.  You can see this on the copying task, where an LSTM is able to ignore the dormant-phase of the sequence well enough to fit the training data, but it hasn’t perfectly learned to keep this phase separate, so it generalizes very poorly if the length of this phase is increased at test time.
>
> [1] Suter et al. 2018, Robustly disentangled causal mechanisms: Validating deep representations for interventional robustness.
> [2] Bengio et. al, 2012, Representation learning: A review and new perspectives.
> [3] Locatello et al., 2018, Challenging common assumptions in the unsupervised learning of disentangled representations.

---

> > ### Author Response · Authors · 2019-11-07
> > **Challenges ML community is facing. (2/2)**
> >
> > “what are the challenges that the ML community is now facing”
> >
> > Current machine learning methods often perform poorly when they are required to generalize beyond the training distribution, which is what is often needed in practice. It is not enough to obtain good generalization on a test set sampled from the same distribution as the training data, we would also like what has been learned in one setting to generalize well in other related distributions. These distributions may involve the same concepts that were seen previously by the learner, with the changes typically arising because of actions of agents. More generally, we would like what has been learned previously to form a rich base from which very fast adaptation to a new but related distribution can take place, i.e., obtain good transfer.
> >
> > “The mathematical descriptions in 2.2, 2.3 and 2.4 are very hard to follow. There is no cost function (for training) around, but there are discussions of  'gradient'. Gradient of which function?”
> >
> > We want to emphasize a very important point, which is that RIMs is a recurrent architecture and it is a drop-in replacement for an LSTM or GRU cell, following the exact same interface with the exact same inputs and outputs.  There is no change to the loss function which results from using RIMs.

---

> ### Author Response · Authors · 2019-11-14
> **Updated Impression ?**
>
> Hello,
>
> R1, We believe, we have addressed your concerns and clarified some of your points.
>
> We hope to have changed your assessment of our work for the better; should that not be the case, please do not hesitate to get in touch with us.
>
> Thanks for your time.

---

> > ### Comment · AnonReviewer1 · 2019-11-14
> > **Thanks for the explanation**
> >
> > The authors have answered several of my comments. I do agree with some intuitions behind this approach. Since the entire field is searching for good philosophy and innovative approaches, trying to get inspirations from other domains such as the causal community is the natural thing to do. But I still think the algorithmic implementations do not necessarily  realize the high-level ideas. The "gap" comment still holds. Nonetheless, I adjusted the score to "weak accept".

---

### Official Review · AnonReviewer3 · 2019-10-18
**Official Blind Review #3**

**Rating:** 6

**Review:**

This paper proposes a neural network architecture consisting of multiple independent recurrent modules that interact sparingly. These independent modules are not all used simultaneously, a subset of them is active at each time step. This subset of active modules is chosen through an attention mechanism. The idea behind this architecture is that it would allow the different modules to specialize in different mechanisms and that would allow compositionality. The empirical results suggest that the proposed approach is able to generalize better than traditional architectures (which all have the implicit assumption that all processes interact).

This paper is well-written and it provides a very thorough empirical analysis of the proposed idea. Because it is not in my area of expertise I’m not confident that I can assess its novelty or its relationship to other existing approaches.

In terms of presentation, I recommend the authors to enlarge some of the figures in the paper (e.g., I can’t read the small box in Figure 1) and to not use citations as nouns (e.g., “The mechanisms of this attention mechanism follow (Vaswani et al., 2017; Santoro et al., 2018), with the …”). I would also like to point out that although fairly different in how they tackle the problem, the work of Arjovsky et al. (2019) seems to be related to this one.

Three questions I believe were not answered in the paper are:

1) How is the performance related to the total number of subsystems (and the number of *active* ones). I can only see results related to that in Table 1, but the variation in the number of modules is pretty small (4-6). The results also don’t give any indication whether we want to have more modules active at each time step, if there’s a sweet spot, etc. It is said that the method seems to be robust to this choice but this claim is made because it performs similarly for the values 5 and 6 if I recall correctly.

2) Is there any incentive in this architecture for a module to not simply “give up”? I mean, the modules are not necessarily incentivized to be used as often as possible, so could it be the case that a module learns to set its weights to zero?

3) Would it make sense to present baseline results for an architecture that uses attention? It seems to me that LSTM was often the baseline of choice but RIMs have two important components: multiple LSTMs and an attention mechanism. Could the attention mechanism be explaining some of the results we are seeing?

Finally, despite the very long appendix, I feel there are important details missing with respect to the empirical setup, at least in the Atari experiments which I’m more familiar with. Was stochasticity used, that is, sticky actions (Machado et al., 2018)? Moreover, for how long was PPO (and RIMs-PPO) trained in terms of number of frames? Finally, I’d recommend the authors to include a table with the actual average (and standard deviation) performance in each Atari games. It is really hard to know how well a method is doing by just squinting at learning curves. It is hard to know if the results are significant without a notion of variance.


References:

Martín Arjovsky, Léon Bottou, Ishaan Gulrajani, David Lopez-Paz: Invariant Risk Minimization. CoRR abs/1907.02893 (2019)

Marlos C. Machado, Marc G. Bellemare, Erik Talvitie, Joel Veness, Matthew J. Hausknecht, Michael Bowling: Revisiting the Arcade Learning Environment: Evaluation Protocols and Open Problems for General Agents. J. Artif. Intell. Res. 61: 523-562 (2018)


------


>>> Update after rebuttal: I stand by my score after the rebuttal.

Unfortunately I'm not an expert in this area and I don't feel confident in having a very strong opinion about this paper, willing to fight for its acceptance. I also agree with concerns raised by other reviewers. As I stated in the discussion with the authors, the clarifications and additional experiment does improve the paper a bit.

**Experience Assessment:**

I do not know much about this area.

**Review Assessment: Checking Correctness Of Derivations And Theory:**

N/A

**Review Assessment: Checking Correctness Of Experiments:**

I assessed the sensibility of the experiments.

**Review Assessment: Thoroughness In Paper Reading:**

I read the paper thoroughly.

---

> ### Author Response · Authors · 2019-11-12
> **R3 Response**
>
> Thank you for your feedback, and we appreciate your comment that “this paper is well-written and it provides a very thorough empirical analysis of the proposed idea”.
>
> “It is said that the method seems to be robust to this choice [of number of activated RIMs] but this claim is made because it performs similarly for the values 5 and 6 if I recall correctly.”
>
> On the larger scale experiments we mostly used 6 RIMs and considered making 4 or 5 activated per-step.  For the smaller scale experiments, we performed a full hyperparameter sweep.  We did find that only keeping one or two activated per step tended to hurt optimization, but 3, 4, and 5 all performed well (when using a total of 6 RIMs).  We also note that using 6 RIMs and a sparsity of k_A = 4 worked well on all of the tasks that we considered, including Atari-PPO, BabyAI, Copying, Sequential MNIST, Mujoco Imitation Learning, and bouncing balls.
>
> Also see the top level comment to all reviewers for a new experiment that explores this more thoroughly.
>
> “2) Is there any incentive in this architecture for a module to not simply “give up”? I mean, the modules are not necessarily incentivized to be used as often as possible, so could it be the case that a module learns to set its weights to zero?”
>
> This is a really interesting question.  In practice we never observed an activation pattern where the model uses the same 3 RIMs on all time steps and has 3 RIMs always dormant (where there are 6 RIMs total).  Nonetheless it’s easy to see that such a solution could fit the data (since a single RIM is as powerful as an LSTM, and an LSTM is usually capable of fitting the data).
>
> One hypothesis is that because deactivated RIMs keep the same state, they do a better job of keeping their values and sustaining gradients over many time steps.  Thus when the model tries to use information from more distant time steps, it naturally looks towards the currently deactivated RIMs, which in turn causes them to be activated on the following steps.
>
> There is also an issue of the model having an incentive to use all the RIMs to complete a complex task, especially if each RIM is relatively simple.
>
> As a side note, the notion of RIMs “giving up” was framed negatively in your question, but we could hypothetically imagine a situation where it becomes a desirable property.  For example if it were the case that some RIMs never activated once the task is fit, we could prune them out and achieve a smaller but functionally identical model, thus allowing the model’s complexity to adaptively adjust to the complexity of the problem.  Note that this is way beyond the scope of this paper, but I think it might be worth considering this angle.
>
> “3) Would it make sense to present baseline results for an architecture that uses attention? It seems to me that LSTM was often the baseline of choice but RIMs have two important components: multiple LSTMs and an attention mechanism. Could the attention mechanism be explaining some of the results we are seeing?”
>
> Many of our baselines do use attention.  For example the relational memory core uses attention between modules and the transformer does as well.  We found that neither matched the generalization performance of RIMs on the tasks we considered (Table 1 and 2).
>
> “ important details missing with respect to the empirical setup, at least in the Atari experiments”
>
> We trained both the baselines as well as proposed method for 40M steps. We updated the paper include a new table (Table 7 in the appendix) with the average as well as standard deviations for our method as well as the baseline.

---

> ### Author Response · Authors · 2019-11-14
> **Updated Impression ?**
>
> Hello,
>
> We believe, we have addressed your concerns and clarified some of your points. Do you have an updated impression of our paper? Thanks for your consideration and time. Appreciate it.

---

> ### Author Response · Authors · 2019-11-14
> **More feedback ?**
>
> Dear Reviewer,
>
> Your feedback has also helped in improving the presentation of the paper. We have done new experiments and analysis related to your questions. Since the review discussion period is going to end, we would appreciate any feedback that you might have. We would be happy to provide further revisions or experiments to address any remaining issues, and would appreciate a response from you on the points that we raised.
>
> Thanks for your time.

---

> > ### Comment · AnonReviewer3 · 2019-11-14
> > **Perception**
> >
> > My questions have been addressed. Thanks for that. I am leaning towards accepting this paper and my score reflects that. With respect to additional details, you never answered my question of whether you used sticky actions or not in the Atari experiments. This is not going to change my assessment of the paper, but it is important for reproducibility and for understand the consequences of the presented results. Without sticky actions it is quite easy to memorize successful trajectories in Atari games.

---

> > > ### Author Response · Authors · 2019-11-14
> > > **Update.**
> > >
> > > Hello Reviewer,
> > >
> > > Thanks for your reply. We agree with you. We apologize for not mentioning. Sticky actions were used.
> > > We'll open source our code.
> > >
> > > Thanks.

---

> > > > ### Author Response · Authors · 2019-11-14
> > > > **Update 2**
> > > >
> > > > Anything else which we can do which can help the reviewer in improving the perception and understanding of the work ?

---

### Official Review · AnonReviewer2 · 2019-10-24
**Official Blind Review #2**

**Rating:** 6

**Review:**

This paper draws inspiration from Physical world and considers an independent mechansim among recurrent modules. The authors apply the proposed RIM to several relatively simple tasks and show some advantages.

In general, I like the idea of making recurrent cells operate with nearly independent transition dynamics and interact only sparingly through the attention bottleneck. It is essentially to combine some environment prior into the model design. It makes senses to me that RIMs will work better in environments that objects and background are nearly independent and only interact with each other when collision happens. RIMs share similar to spirits with capsule networks, and its recurrent cells serve somewhat similar role to capsules. Such independent mechanism, selective activation and sparse communication is very inspiring and is indeed a potentially very useful way of modeling the physical world.

For the model itself, I appreciate its simplicity, but I also have some concerns.

1) For the selective activation of RIMs, the number of activated RIMS is a hyperparameter and needs to be pre-defined. According to your experiments, I believe you need tune this hyperparameter a little bit in order to obtain the best performance. First of all, the design of a fixed number of activated RIMs does not seem to be reasonable and is also highly dependent on your task. I believe the framework will be more interesting if the model can determine this number automatically.

2) I find it quite interesting that the top-down attention in selective RIM activation is corresponding to the states of these recurrent cells. I am wondering what if you do not select these top K activation and directly train it using the entire distribution of the soft attention output?

For the experiments, I think they all serve the purpose of showing the advanatges of RIMs quite well, except that thery are relatively easy task. However, it is still interesting to see that RIMs obtain significant gains over some baselines. Some of the details can be made more clear, such as loss function and evaluation metrics in every task. It is sometimes difficult to find what loss function you are using. I suggest the authors make the experiments more self-contained in the main paper, such that authors do not need frequently scroll down to the appendix and check the details.

**Experience Assessment:**

I do not know much about this area.

**Review Assessment: Checking Correctness Of Derivations And Theory:**

N/A

**Review Assessment: Checking Correctness Of Experiments:**

I assessed the sensibility of the experiments.

**Review Assessment: Thoroughness In Paper Reading:**

I read the paper at least twice and used my best judgement in assessing the paper.

---

> ### Author Response · Authors · 2019-11-12
> **R2 Response**
>
> Thank you for your review.  We appreciate that you like the idea and believe that it is “very inspiring and is indeed a potentially very useful way of modeling the physical world”.
>
> “1) For the selective activation of RIMs, the number of activated RIMS is a hyperparameter and needs to be pre-defined. According to your experiments, I believe you need tune this hyperparameter a little bit in order to obtain the best performance. First of all, the design of a fixed number of activated RIMs does not seem to be reasonable and is also highly dependent on your task.”
>
> Our experiments show that this hyperparameter is not very task dependent.  For example, having 6 RIMs with 4 RIMs activated-per-step actually works well on all the tasks we considered.  Intuitively, because the model learns with a certain number of RIMs activated-per-step, it is able to adapt the information that it puts on RIMs to handle that level of sparsity.
>
> As a side note, many papers that have considered sparse attention such as Outrageously Large Neural Networks (https://arxiv.org/abs/1701.06538), Sparse Attentive Backtracking (https://arxiv.org/abs/1809.03702) and Sparse Transformers (https://openreview.net/pdf?id=Hye87grYDH) achieved robustly good results using a fixed top-k sparsity as a hyperparameter.
>
> We ran an experiment (see the top level comment aimed at all reviewers) where we use a much larger number of RIMs and verify that the amount of RIM sparsity is still very flexible.
>
> “2) I find it quite interesting that the top-down attention in selective RIM activation is corresponding to the states of these recurrent cells. I am wondering what if you do not select these top K activation and directly train it using the entire distribution of the soft attention output?”
>
> If I understand correctly, you’re asking about removing the selective activation mechanism and updating all RIMs on all time steps?  We studied this extensively and found that on the hard generalization tasks, RIMs performs substantially worse without selective activation.  It also generally hurts results on Atari.
>
> “For the experiments, I think they all serve the purpose of showing the advantages of RIMs quite well, except that they are relatively easy tasks”
>
> Our aim has been to show that RIMs is an idea with breadth—over the quite disparate domains of sequence memorization, sequence classification tasks, control tasks, robustness to distractions and prediction tasks. More broadly, RIMs is a step towards addressing an important issue in deep learning: how to build robust models given that internal state spaces are continuous, high dimensional, and often unbounded. The way RIMs do is by modularizing the dynamics model. The high level intuition is, this form of computational modularity is essential for modeling independent mechanisms in a way that generalizes, but it is not clear how this is implemented, or what learning mechanisms may lead to such distributed representations. Here, we investigate this question by using recurrent neural networks with modular interactions, and evaluate it on various different tasks. We find that this modular inductive bias has distinct advantages: (1) it learns distributed representations in subsystems with limited interactions, consistent with observations in the brain (2) it shows improved generalization performance and transferability to novel environments.

---

> ### Author Response · Authors · 2019-11-14
> **Updated Impression ?**
>
> Hello,
>
> We believe, we have addressed your concerns and clarified some of your points. Do you have an updated impression of our paper?  Should that not be the case, please do not hesitate to get in touch with us. Thanks for your consideration and time. Appreciate it.

---

### Author Response · Authors · 2019-11-12
**New Experiment on Activation Sparsity hyperparameter (especially R2 and R3).**

Several reviewers expressed concern about the “RIM sparsity hyperparameter” which we call k_A.  In practice we’ve found that it’s fairly resilient across tasks and number of RIMs.  For example, using 6 RIMs and a sparsity of k_A = 4 worked well on all of the tasks that we considered, including Atari-PPO, BabyAI, Copying, Sequential MNIST, Mujoco Imitation Learning, and bouncing balls.

Nonetheless we believe that understanding the role of this hyperparameter more carefully is important so we conducted a new analysis where we study the role of the RIM sparsity hyperparameter as we grow the number of RIMs.

We analyzed this on the copying task, considering 9 RIMs, 16 RIMs, and 24 RIMs.  Using 9 RIMs, we obtained ideal performance with k_A from 3 to 6.  Using 16 RIMs, we obtained ideal performance with k_A from 4 to 12.  Using 24 RIMs, we obtained ideal performance with k_A between 6 and 20.  As percentages, this is 33% to 66% for 9 RIMs, 25% to 75% for 16 RIMs, and 25% to 83% for 24 RIMs.

k_A Sparsity / Total RIMs: Test Loss (Train Loss).
2/9: 0.15 (0.24)
3/9: 0.02 (0.03)
4/9: 0.00 (0.01)
5/9: 0.00 (0.01)
6/9: 0.01 (0.01)
7/9: 0.07 (0.04)
8/9: 0.10 (0.00)
9/9: 0.24 (0.03)

4/16: 0.02 (0.05)
6/16: 0.00 (0.00)
8/16: 0.00 (0.00)
10/16: 0.00 (0.00)
12/16: 0.00 (0.00)
14/16: 0.26 (0.01)

6/24: 0.00 (0.01)
8/24: 0.00 (0.00)
16/24: 0.00 (0.00)
20/24: 0.00 (0.00)
22/24: 0.42 (0.00)

LSTM: 2.28 (0.0)

----

It’s worth noting here that the situation is somewhat analogous to setting the rate hyperparameter for the dropout regularizer.  p=0.5 is often quite good across a variety of tasks.  Using a very large p, such as p=0.9 may lead to underfitting and using too small p, such as p=0.1 may show a reduced benefit for generalization.  Indeed this is exactly what we observe with RIMs.

---

### Decision · Program_Chairs · 2019-12-19

**Decision:**

Reject

**Comment:**

This paper has, at its core, a potential for constituting a valuable contribution. However, there was a shared belief among reviewers (that I also share) that the paper still has much room for improvement in terms of presentation and justification of the claims. I hope that the authors will be able to address the feedback they received to make this submission get where it should be.